# TALPID3 controls centrosome and cell polarity and the human ortholog *KIAA0586* is mutated in Joubert syndrome (*JBTS23*)

Louise A Stephen[1], Hasan Tawamie[2], Gemma M Davis[1], Lars Tebbe[3], Peter Nürnberg[4,5], Gudrun Nürnberg[4], Holger Thiele[4], Michaela Thoenes[6], Eugen Boltshauser[7], Steffen Uebe[2], Oliver Rompel[8], André Reis[2], Arif B Ekici[2], Lynn McTeir[1], Amy M Fraser[1], Emma A Hall[9], Pleasantine Mill[9], Nicolas Daudet[10], Courtney Cross[11], Uwe Wolfrum[3], Rami Abou Jamra[2,12,13†], Megan G Davey[1*†], Hanno J Bolz[6,14*†]

[1]Division of Developmental Biology, The Roslin Institute, University of Edinburgh, Edinburgh, United Kingdom; [2]Institute of Human Genetics, Friedrich-Alexander-Universität Erlangen-Nürnberg, Erlangen, Germany; [3]Cell and Matrix Biology, Institute of Zoology, Johannes Gutenberg University of Mainz, Mainz, Germany; [4]Cologne Center for Genomics, Center for Molecular Medicine Cologne, University of Cologne, Cologne, Germany; [5]Cologne Cluster of Excellence, University of Cologne, Cologne, Germany; [6]Institute of Human Genetics, University Hospital of Cologne, Cologne, Germany; [7]Department of Paediatric Neurology, University Children's Hospital Zurich, Zurich, Switzerland; [8]Institute of Radiology, Friedrich-Alexander-Universität Erlangen-Nürnberg, Erlangen, Germany; [9]Medical Research Council Human Genetics Unit, MRC Institute of Genetics and Molecular Medicine, University of Edinburgh, Edinburgh, United Kingdom; [10]UCL Ear Institute, University College London, London, United Kingdom; [11]School of Osteopathic Medicine, A.T. Still University, Mesa, United States; [12]Centogene, Rostock, Germany; [13]Institute of Human Genetics, Leipzig University, Leipzig, Germany; [14]Bioscientia Center for Human Genetics, Bioscientia International Business, Ingelheim am Rhein, Germany

*For correspondence: megan.davey@roslin.ed.ac.uk (MGD); hanno.bolz@uk-koeln.de (HJB)

†These authors contributed equally to this work

Competing interests: The authors declare that no competing interests exist.

**Abstract** Joubert syndrome (JBTS) is a severe recessive neurodevelopmental ciliopathy which can affect several organ systems. Mutations in known JBTS genes account for approximately half of the cases. By homozygosity mapping and whole-exome sequencing, we identified a novel locus, *JBTS23*, with a homozygous splice site mutation in *KIAA0586* (alias *TALPID3*), a known lethal ciliopathy locus in model organisms. Truncating *KIAA0586* mutations were identified in two additional patients with JBTS. One mutation, c.428delG (p.Arg143Lysfs*4), is unexpectedly common in the general population and may be a major contributor to JBTS. We demonstrate KIAA0586 protein localization at the basal body in human and mouse photoreceptors, as is common for JBTS proteins, and also in pericentriolar locations. We show that loss of TALPID3 (KIAA0586) function in animal models causes abnormal tissue polarity, centrosome length and orientation, and centriolar satellites. We propose that JBTS and other ciliopathies may in part result from cell polarity defects.

## Introduction

Joubert syndrome (JBTS) is a rare ciliopathy characterized by a specific midhindbrain malformation presenting as 'molar tooth sign' on axial MRI. Patients typically have a perturbed respiratory pattern in the neonatal period and pronounced psychomotor delay. Depending on the genetic subtype, there

**eLife digest** Joubert syndrome is a rare and severe neurodevelopmental disease in which two parts of the brain called the cerebellar vermis and brainstem do not develop properly. The disease is caused by defects in the formation of small projections from the surface of cells, called cilia, which are essential for signalling processes inside cells. Mutations in at least 25 genes are known to cause Joubert syndrome, and all encode proteins that create or maintain cilia. However, these mutations account for only half of the cases that have been studied, which indicates that mutations in other genes may also cause Joubert syndrome.

Here, Stephen et al. used genetic techniques called 'homozygosity mapping' and 'whole-exome sequencing' to search for other mutations that might cause the disease. They found that mutations in a gene encoding a protein called KIAA0586 also cause Joubert syndrome in humans. One of these mutations (c.428delG) is unexpectedly common in the healthy human population. It might be a major contributor to Joubert syndrome, and the manifestation of Joubert syndrome in individuals with this mutation might depend on the presence and nature of other mutations in KIAA0586 and in other genes.

The TALPID3 protein in chickens and other 'model' animals is the equivalent of human KIAA0586. A loss of TALPID3 protein in animals has been shown to stop cilia from forming. This protein is found in a structure called the basal body, which is part of a larger structure called the centrosome that anchors cilia to the cell. Here, Stephen et al. show that this is also true in mouse and human eye cells.

Further experiments using chicken embryos show that a loss of the TALPID3 protein alters the location of centrosomes inside cells. TALPID3 is also required for cells and organs to develop the correct polarity, that is, directional differences in their structure and shape. The centrosomes of chicken brain cells that lacked TALPID3 were poorly positioned at the cell surface and abnormally long, which is likely responsible for the cilia failing to form.

Stephen et al.'s findings suggest that KIAA0586 is also important for human development through its ability to control the centrosome. Defects in TALPID3 have a more severe effect on animal models than many of the identified KIAA0586 mutations have on humans. Therefore, the next step in this research is to find a more suitable animal in which to study the role of this protein, which may inform efforts to develop treatments for Joubert syndrome.

may be additional retinal degeneration, nephronophthisis, liver fibrosis, and skeletal abnormalities (such as polydactyly). JBTS is genetically heterogeneous, with recessive mutations reported in more than 20 genes encoding proteins related to the function of cilia and associated structures (*Romani et al., 2013*; *Bachmann-Gagescu et al., 2015*).

Cilia are axoneme-based organelles which protrude into the extracellular milieu, anchored to the cell by a modified centriole (basal body). They are present in virtually every cell type (*Christensen et al., 2007*). Non-motile 'primary' cilia play essential roles in mechanotransduction, chemosensation, and intracellular signal transduction, including Hedgehog (Hh), PDGFα, and WNT pathways, in embryonic development and adult tissue homeostasis (*Goetz and Anderson, 2010*). In addition, highly modified and specialized cilia constitute the light-sensitive outer segments of retinal photoreceptor cells. Dysfunction of cilia, centrioles of basal bodies, and centrosomes can lead to a spectrum of developmental single- or multi-organ disorders termed 'ciliopathies' (*Bettencourt-Dias et al., 2011*).

KIAA0586 (TALPID3; MIM #610178, MIM #000979-9031) is essential for vertebrate development and ciliogenesis. The KIAA0586 (TALPID3) protein is localized at the centrosome in human, chicken, mouse, and zebrafish cells (*Yin et al., 2009*; *Ben et al., 2011*; *Wu et al., 2014*), and in particular, at the distal end of the mother centriole—the basal body of cilia (*Kobayashi et al., 2014a*). In model organisms, KIAA0586 null mutations cause failure of basal body docking and loss of cilia, leading to early embryonic lethal phenotypes (*Davey et al., 2006*; *Bangs et al., 2011*; *Ben et al., 2011*; *Stephen et al., 2013*). KIAA0586 (TALPID3) binding partners include PCM1, Cep120, and CP110, which interact with a known JBTS protein, CEP290 (*Tsang and Dynlacht, 2013*).

Here, we report three JBTS families with loss-of-function mutations in KIAA0586. Using animal models, we demonstrate that TALPID3 (KIAA0586) is not only essential for transduction of Hedgehog signaling but plays an important role in centrosomal localization, orientation, and length. Finally, and beyond its established requirement for ciliogenesis, TALPID3 (KIAA0586) plays a key role in cell and tissue polarity.

## Results

### Clinical description of patients with *KIAA0586*-associated JBTS

The diagnosis of JBTS was based on the presence of a molar tooth sign in all three families.

Family 1 (*Figure 1A*) is a consanguineous Kurdish family from northeast Syria. The two affected siblings were examined at the age of 6 years and 10 months (MR026-01) and 2 years and 2 months (MR026-04), respectively. Pregnancy, delivery, and birth parameters of both children were unremarkable. In the neonatal period, both were hypotonic and weepy. Motor and speech development in MR026-01 were delayed, and his IQ was estimated to be between 50 and 70. Further symptoms were severe myopia, scoliosis, brachydactyly, distinct facial characteristics, and recurrent febrile seizures. Height was reduced (108 cm, −2.6 SD), weight was normal (22 kg, −0.27 SD), and head circumference was increased (57 cm, +2.3 SD). MR026-04 had not reached any milestones, and at the age of 7 years, she was wheelchair-bound. Cognitive abilities were weaker than in her brother, with an IQ estimated to be below 35. MR026-04 had similar physical characteristics as her brother, severe muscular hypotonia, prolonged and therapy-resistant seizures since the age of 14 months, and hypothyroidism. At the time of examination, her height was 91 cm (1 SD), weight was 11.5 kg (−0.7 SD), and there was macrocephaly (head circumference of 59 cm, +8 SD).

Family 2 (*Figure 1B*) is of North American origin. Patient MD1 was born at 34 3/7 weeks gestation following preterm premature rupture of membranes at 26 weeks. At birth, patient MD1 was found to have cardiac defects including a patent ductus arteriosus (PDA), patent foramen ovale (PFO) and a 3/6 ventricular septal defect (VSD) causing persistent pulmonary hypertension 24 hr after birth. The PDA and PFO resolved, and VSD was at 2/6 within 22 days. At 7 months, MD1 was found to have a superior vena cava duplication. At 2 years of age, MD1 had hypotonia which inhibited motor actions, although she crawled proficiently, used sign language and single words, and self-fed by hand and with utensils. In addition, she had type I bilateral Duane syndrome with no abduction in either eye, narrowing of the palprebal fissure of the inturned eye, was farsighted, had thin tooth enamel, held her jaw sideways in a cross-bite pattern, and had long fingers with a slight clinodactyly of the fifth finger. She had a broad forehead, arched eyebrows, ptosis of the right eye, and a triangle-shaped mouth. Her receptive language was good. There was intermittent hyperpnea/apnea during awake periods. Patient MD1 had no liver, kidney, or eye abnormalities at 2 years of age.

Family 3 (*Figure 1C*) is of German origin: patient G2 displayed a relatively mild JBTS phenotype with developmental delay and behavioral abnormalities, but no dysmorphic signs and no renal, retinal, skeletal, or liver abnormalities. His symptoms were described previously (*Figure 1C*, *Dafinger et al., 2011*).

### Mutations of *KIAA0586* cause JBTS

We have identified *KIAA0586* mutations in three JBTS families (*Figure 1A–D*). Genome-wide SNP genotyping in Family 1 identified eight homozygous chromosomal candidate regions with a total range of 67.1 Mb. By WES, the homozygous mutation c.2414-1G>C in intron 17, affecting the invariant consensus of the exon 18 acceptor splice site, was found in the index patient, MR026-01, and his affected sister, MR026-04. Segregation analysis in the family was compatible with causality (*Figure 1A*). The mutation was absent from 372 healthy Syrian controls, including 92 of Kurdish origin, and not listed in the ExAC database.

In patient MD1 from Family 2, WES identified compound heterozygosity for the *KIAA0586* mutations c.428delG (p.Arg143Lysfs*4; rs534542684; MAF of 0.39% in ExAC db) and c.2512C>T (p.Arg838*), each inherited from a healthy parent (*Figure 1B*), and both resulting in premature stop codons. Because the coiled-coil domain, which is essential for KIAA0586 function in mouse, chicken, and zebrafish (residues 531–571 and residues 497–530 in human and chicken KIAA0586 (TALPID3), respectively; *Figure 1F*), would be lost in a truncated protein derived from the c.428delG mutation, we consider it a loss-of-function mutation (as is the case for the *talpid³* chicken mutation which introduces a frameshift 3′ to c.428 in the chicken ortholog, *Figure 1F*). Like the *Talpid3/TALPID3* null mutations in mouse and chicken, c.428delG is clearly recessive because the father of the patient is a healthy carrier. The c.2512C>T (p.Arg838*) mutation is predicted to lead to nonsense-mediated decay (NMD) or a truncated protein, but with preservation of the essential coiled-coil domain.

The simplex patient of Family 3, G2, was a known carrier of a heterozygous N-terminal frameshift mutation in exon 3 of the *JBTS12* gene *KIF7*, c.811delG (p.Glu271Argfs*51) (*Dafinger et al., 2011*).

WES of the family trio (patient G2 and his parents) additionally identified the c.428delG (p.Arg143Lysfs*4) mutation in *KIAA0586*, in the patient (*Figure 1C*). We hypothesized that disease in patient G2 could be due to biallelic mutations either in *KIF7* (*JBTS12*) or in *KIAA0586* (*JBTS23*), assuming that the 'missing mutation' has escaped detection by sequencing due to an extra-exonic localization. Genome-wide CGH (Affymetrix 6.0 SNP array) did not reveal structural alterations adjacent to or within *KIF7* (*Dafinger et al., 2011*) or *KIAA0586*, thereby largely excluding a large deletion or duplication. PCR amplification and subsequent sequencing of *KIAA0586* exons from cDNA did not reveal aberrant splicing as a potential hint for a deep intronic splice site mutation. Because *KIF7* and *KIAA0586* both encode modulators of GLI processing and c.428delG$_{KIAA0586}$ and c.811delG$_{KIF7}$ likely represent recessive loss-of-function mutations, we investigated the possibility of a potential epistatic effect predisposing to JBTS. No such interactions were identified in mouse and chicken experiments (details are fully described in *Figure 1—figure supplement 1*). Therefore, unidentified mutations are likely to be involved, either mutations in *KIF7*, *KIAA0586* (e.g., deep intronic mutations or alterations in non-coding regulatory regions which would both not be covered by WES) or biallelic mutations in another (yet unknown) JBTS gene. WES revealed further heterozygous missense variants in three known recessive ciliopathy genes in patient G2 (*Figure 1C*), all affecting evolutionarily conserved residues of the respective proteins: (1) c.536 G>A (p.Arg179His, rs140259402; MAF of 0.001647% in ExAC db) in *CEP41*, the gene associated with *JBTS15* (*Lee et al., 2012*). (2) c.3181A>G (p.Ile1061Val; MAF of 0.01155% in ExAC db) in *KIF14*, a gene associated with a lethal fetal ciliopathy phenotype (*Filges et al., 2014*). (3) c.1333G>C (p.Ala445Pro, rs61734466; MAF of 0.6609% in ExAC db) in *WDPCP*, the gene associated with Bardet–Biedl syndrome type 15 (*BBS15*), and a putative contributor to Meckel Gruber syndrome (*Kim et al., 2010*). All variants were of paternal origin and rare in the general population except the *WDPCP* allele, which had been maternally inherited and which has been annotated homozygously in five healthy individuals (ExAC db), indicating that this is a benign variant. Genome-wide CGH (Affymetrix 6.0 SNP array) did not show structural alterations adjacent to or within *CEP41*, *KIF14*, or *WDPCP*. In addition, we searched the WES data of patient G2 for heterozygous putative loss-of-function (that is, truncating) variants in genes with documented ciliary function. This revealed a paternally inherited frameshift variant, c.206_207insA (p.Ser70Valfs*3), in *PLA2G3*, the gene encoding phospholipase A2. In a functional genomic screen, PLA2G3 was found to be a negative regulator of ciliogenesis and ciliary membrane protein targeting (*Kim et al., 2010*). The p.Ser70Valfs*3$_{PLA2G3}$ variant is relatively common, but has not been documented in homozygous state in healthy individuals (MAF of 0.4060 in ExAC db).

We also filtered for known JBTS genes carrying at least two rare variants in patient G2, but we did not identify such a constellation. When applying this to all genes captured in the WES approach, there was also no potentially causative double heterozygosity in a gene of known or probable ciliary function. Filtering for homozygous rare and likely pathogenic variants was negative, compatible with lack of consanguinity in the parents of patient G2.

## Consequences of the *KIAA0586* mutation c.2414-1G>C on mRNA level

The c.2414-1G>C mutation affects the invariant consensus of the acceptor splice site of exon 18. RT-PCR and Sanger sequencing of the fragments amplified from cDNA revealed three aberrant splicing products due to usage of alternative exonic acceptor splice sites at AG motifs within exon 18 and due to skipping of exon 18 (*Figure 1D,E*): a 13-bp deletion that results in a premature termination codon (alternative acceptor splice site at c.2425/2426AG), a 108-bp in-frame deletion (alternative acceptor splice site at c.2520/2521AG), and a 188-bp deletion due to skipping of exon 18 that results in a premature stop codon. These aberrant transcripts were present in the cDNA from both patients, but not in the cDNA of a healthy control individual (*Figure 1D*). The mutant mRNA molecules are likely to be degraded by NMD. If the mutant transcripts were stable, the essential coiled-coil domain (*Figure 1E*), which mediates centrosomal localization and function of KIAA0586 protein (*Yin et al., 2009*; *Wu et al., 2014*), would be preserved.

## KIAA0586 localizes to the basal body of cultured cells and photoreceptor cells of human and mouse retina

KIAA0586 (Talpid3) is a centrosomal protein and localizes to the basal body and the adjacent centriole of primary cilia in human RPE1, IMCD3 cells (*Figure 2A*), and other cell types (*Kobayashi et al., 2014a*;

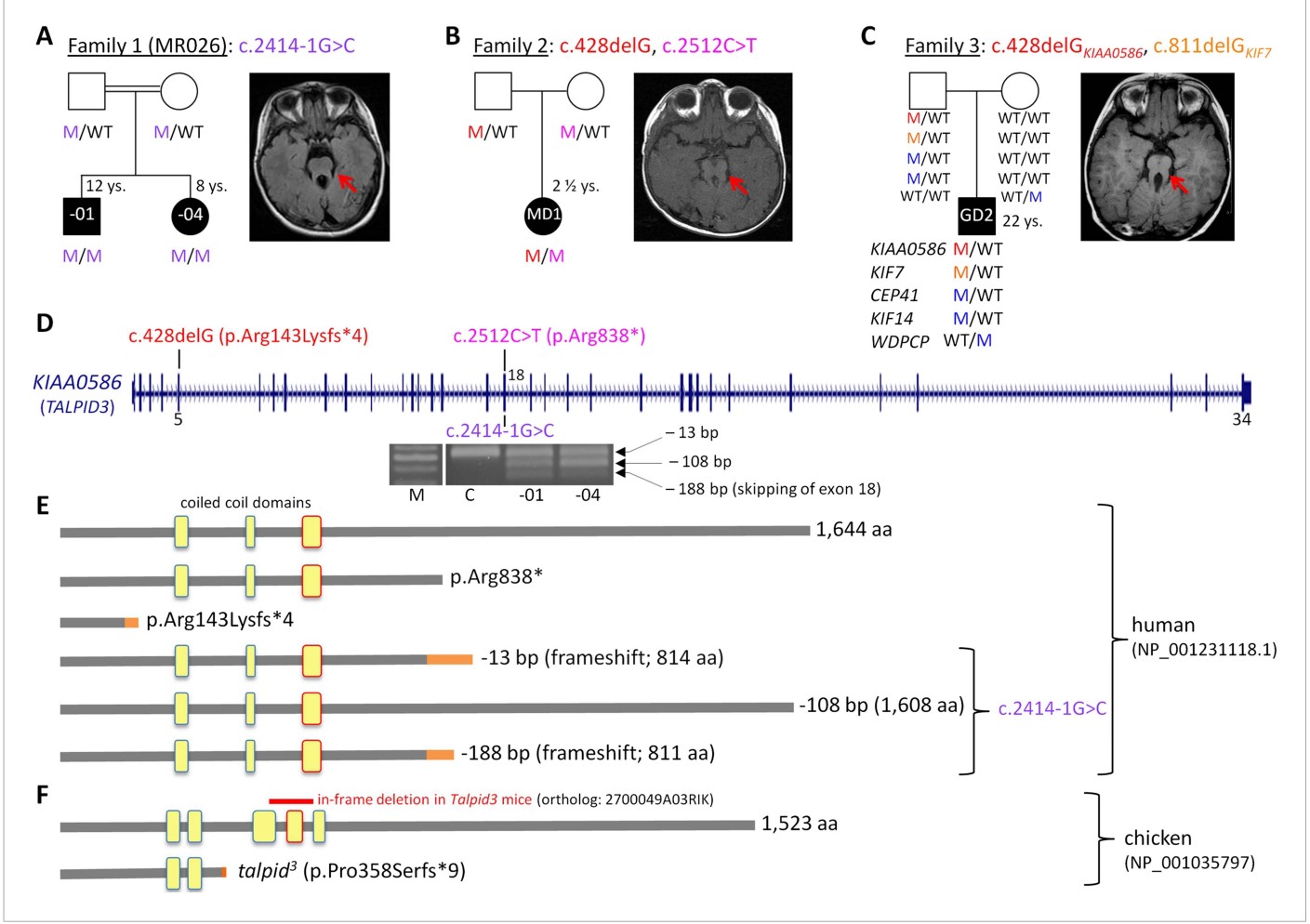

**Figure 1.** Patients with Joubert syndrome (JBTS) and *KIAA0586* mutations (**A–C**). (WT, wildtype; M, mutation). The 'molar tooth sign' in cranial axial MRI is indicated by arrows. (**A**) Family 1: Homozygosity mapping yielded eight homozygous chromosomal candidate regions (not shown), including the *JBTS23* locus comprising *KIAA0586*. Patients MR026-01 and MR026-04 carry a homozygous splice site mutation, c.2414-1G>C. (**B**) Patient MD1 of Family 2 is compound heterozygous for two truncating mutations, including the prevalent c.428delG (p.Arg143Lysfs*4) allele. (**C**) Family 3: Patient G2 is double heterozygous for c.428delG in *KIAA0586*, and a frameshift mutation in *KIF7* (*JBTS12*; c.811delG, p.Glu271Argfs*51). He also carries three potentially pathogenic variants in the ciliopathy genes *CEP41*, *KIF14*, and *WDPCP* (blue). (**D**) Genomic structure of *KIAA0586* with mutations in exons 5 and in/adjacent to exon 18 indicated. The gel electrophoresis shows the aberrant transcripts due to c.2414-1G>C. (**E**) Scheme of human KIAA0586 protein and predicted consequences of JBTS-associated mutations. Orange color: unrelated residues included due to frameshift mutations. The third coiled-coil domain is the counterpart of the functionally essential fourth coiled-coil domain in chicken (framed in red). (**F**) Chicken TALPID3 (KIAA0586) is highly similar to the human protein. The *talpid³* mutation results in an early frameshift and loss of three coiled-coil domains, including the fourth one. The in-frame deletion of exons 11 and 12 of mouse *KIAA0586* (*2700049A03Rik*) is depicted above the scheme of the chicken ortholog.

The following figure supplement is available for figure 1:

**Figure supplement 1.** Analysis of potential interactions between *Talpid3/TALPID3*, *Kif7/KIF7* and *IFT57* in the mouse and in chicken.

---

*Wu et al., 2014*). Immunofluorescence analysis of the retina of *wildtype* C57BL/6 mice allowed us to allocate Talpid3 expression to different retinal layers, namely the photoreceptor layer, the outer and inner plexiform layer, and the ganglion cell layer (*Figure 2B*). Co-staining with the ciliary marker centrin-3 (*Trojan et al., 2008*) demonstrated Talpid3/KIAA0586 localization in the ciliary region at the joint between the inner and outer segment of photoreceptor cells in cryosections through the mouse retina and the retina of a human donor eye (*Figure 2B,C,E*). Higher magnification revealed that Talpid3/KIAA0586 specifically localized at the basal body (mother centriole) and the adjacent centriole as well as between the two centrioles, but not in the connecting cilium of mouse and human

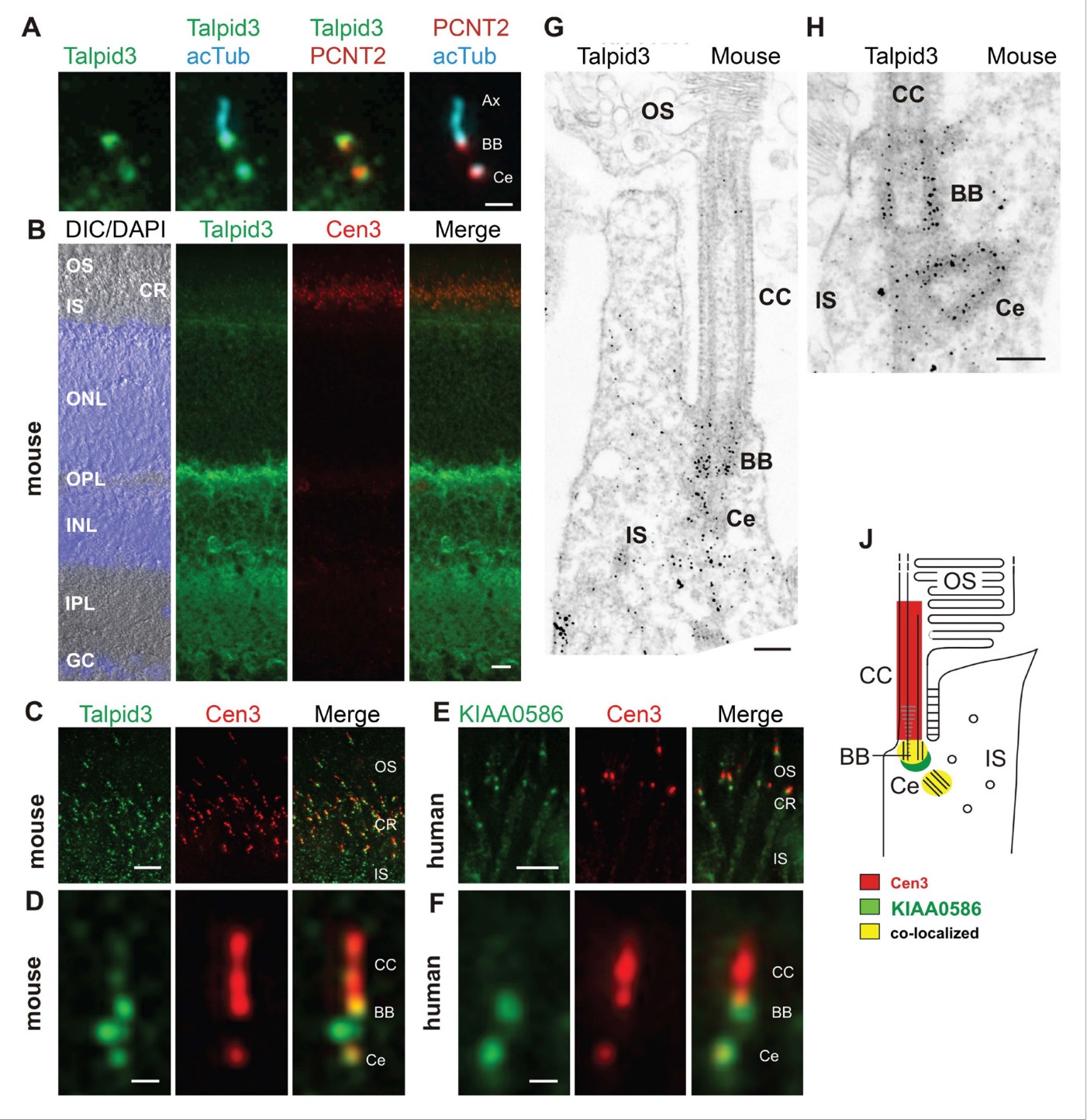

**Figure 2.** Localization of KIAA0586/Talpid3 in primary cilia and in photoreceptor cilia of mammalian retinas. (**A**) Triple labeling of a ciliated IMCD3 cell demonstrates localization of Talpid3 (green) in the basal body (BB) and the adjacent centriole (Ce) at the base of the primary cilium co-stained by antibodies against Pericentrin-2 (PCNT2, red) and anti-acetylated tubulin (acTub, cyan), a biomarker of the axoneme (Ax). (**B**) Longitudinal cryosections through a mouse retina stained for Talpid3 (green) and counterstained for the ciliary marker Centrin-3 (Cen3, red) and for the nuclear DNA marker DAPI reveal Talpid3 localization in the ciliary region (CR) at the joint between the inner (IS) and the outer segment (OS) of the photoreceptor layer, the outer (OPL) and inner plexiform layer (IPL). Overlay of DIC (differential interference contrast) image with DAPI (blue) nuclear stain in the outer (ONL) and the inner nuclear layer (INL) and in the ganglion cell layer (GC). (**C–F**) Immunostaining of cryosections through the photoreceptor layer of a mouse (**C**) and a human retina (**E**) demonstrates co-localization of KIAA0586/Talpid3 and Cen3 in the CR of photoreceptor cells. Higher magnification of double-labeled mouse (**D**) and human (**F**) photoreceptor cilium reveals substantial localization of Talpid3/KIAA0586 at the centriole (Ce), the BB and between the Ce and
*Figure 2 continued on next page*

*Figure 2 Continued*

BB of the photoreceptor cilium, but not in the connecting cilium (CC). (**G**, **H**) Immunoelectron microscopy analysis of longitudinal section through the cilium of a mouse rod photoreceptor cell and (**G**) higher magnification of the ciliary base (**H**) labeled for Talpid3 reveals Talpid3 in the periciliary region namely in the Ce and BB. (**J**) Schematic representation of Talpid3/KIAA0586 localization in the photoreceptor cilium. Scale bars: **A**, 1 µm; **B**, 10 µm; **C**, **E**, 5 µm; **D**, **F**, 0.5 µm; **G**, **H**, 200 nm.

photoreceptor cells (*Figure 2D,F*). These findings were confirmed by immunoelectron microscopy of Talpid3 labeling on sections through mouse photoreceptor cilia (*Figure 2G,H*). Immunostaining was found at centrioles and in the pericentriolar region in the apical inner segment of photoreceptor cells. The spatial distribution of Talpid3/KIAA0586 labeling at the ciliary base of photoreceptor cells is summarized in the scheme of *Figure 2J*.

## Loss of *TALPID3* (*KIAA0586*) causes abnormal tissue and cell polarity

The talpid[3]/Talpid3[−/−] phenotype in model animals has thus far been attributed to the role of TALPID3 in ciliogenesis and the subsequent loss of Hh-dependent patterning. However, the patients in this study did not display any overt defects typical for impaired Hh signaling such as polydactyly or hypotelorism, which have been described in other patients with JBTS (*Bachmann-Gagescu et al., 2015*). Talpid[3] chicken embryos also have polycystic kidneys (*Yin et al., 2009*), a phenotype that is frequently ascribed to a loss of oriented cell division (*Happe et al., 2011*; *Carroll and Yu, 2012*), as well as cell migration defects (*Bangs et al., 2011*), which may also occur due to loss of cell polarity (*Happe et al., 2011*; *Carroll and Yu, 2012*). To investigate if tissue and cell polarity is impaired by a loss of TALPID3 function, we first examined the patterning of the skin and the inner ear, two highly polarized tissues independent of Hh signaling. At E10, embryonic chicken feather buds express β-catenin in an oriented manner, with a larger domain in the anterior part of the bud (*Figure 3A,C*). While 88% of *wildtype* feather buds at E10 are oriented in this manner (n = 117/133), only 21% of stage-matched *talpid[3]* feather buds were (n = 38/179). 22% of *talpid[3]* buds were oriented in the wrong direction (n = 39/179) and 57% had failed to show any orientation of β-catenin expression (n = 102/179; *Figure 3C′*). Talpid[3] feather buds also frequently merged (29% of buds; asterisk, *Figure 3B′′*). Thus, the skin of *talpid[3]* embryos did not show the characteristic rostral-caudal polarization of *wildtype* skin. The hair cells (HCs) of the inner ear (known as the basilar papilla (BP) in chicken) have a highly polarized structure determined by the non-canonical Wnt-PCP signaling pathway. In the *wildtype* chicken, as in mouse, individual HCs exhibit an orientated actin-based stereocilia bundle, the apex of which lies at the abneural side of the cell, where within an actin-free 'bare zone', a single kinocilium (a microtubule-based true cilium) forms (arrow, *Figure 3D*). HCs are frequently used to assess how cell polarity and ciliogenesis are perturbed in mouse mutants (*Goetz and Anderson, 2010*). The HCs of *talpid[3]* embryos formed actin filament bundles (curved line, *Figure 3E*), but no kinocilium, demonstrating that, as with other tissues studied, loss of TALPID3 impairs ciliogenesis. Furthermore, although stereocilia were present in *talpid[3]* HCs, stereocilia bundles frequently lacked polarity compared to *wildtype* HCs as indicated by either centrally located stereocilia bundles in SEM or actin filaments throughout the cell (*talpid[3]* n = 1086/1195; *wt* n = 258/502; *Figure 3D–G,L*). Orientation of the polarized stereocilia bundles that did form in *talpid[3]* HCs was also abnormal (*Figure 3E,G,N*). The orientation of stereocilia was determined in relation to their position to the abneural side of the BP (*Figure 3F,G,M*). 73% of stereocilia of *wildtype* cells (n = 244) were oriented within 40° of the expected angle (90°, compared to 38% of *talpid[3]* cells (n = 237; *Figure 3M′*)). Thus, *talpid[3]* HCs showed disrupted polarity.

## Loss of TALPID3 function causes abnormal intracellular organization

Loss of TALPID3 function prevents basal body docking (*Yin et al., 2009*), which we have previously suggested to be due to failure of centrosome migration (*Stephen et al., 2013*). The migration and subsequent localization and docking of the centriole is crucial to establish polarity and placement of the actin bundle in the HC (*Tarchini et al., 2013*), and we therefore hypothesized that disturbed cell polarity may result from defective centrosome migration in *talpid[3]* HCs. Using antibodies against γ tubulin to determine the localization of the centriole within the actin-negative abneural bare zone in the HCs, 95% of *wildtype* HCs exhibited a basal body (centrosome) within the abneural bare zone

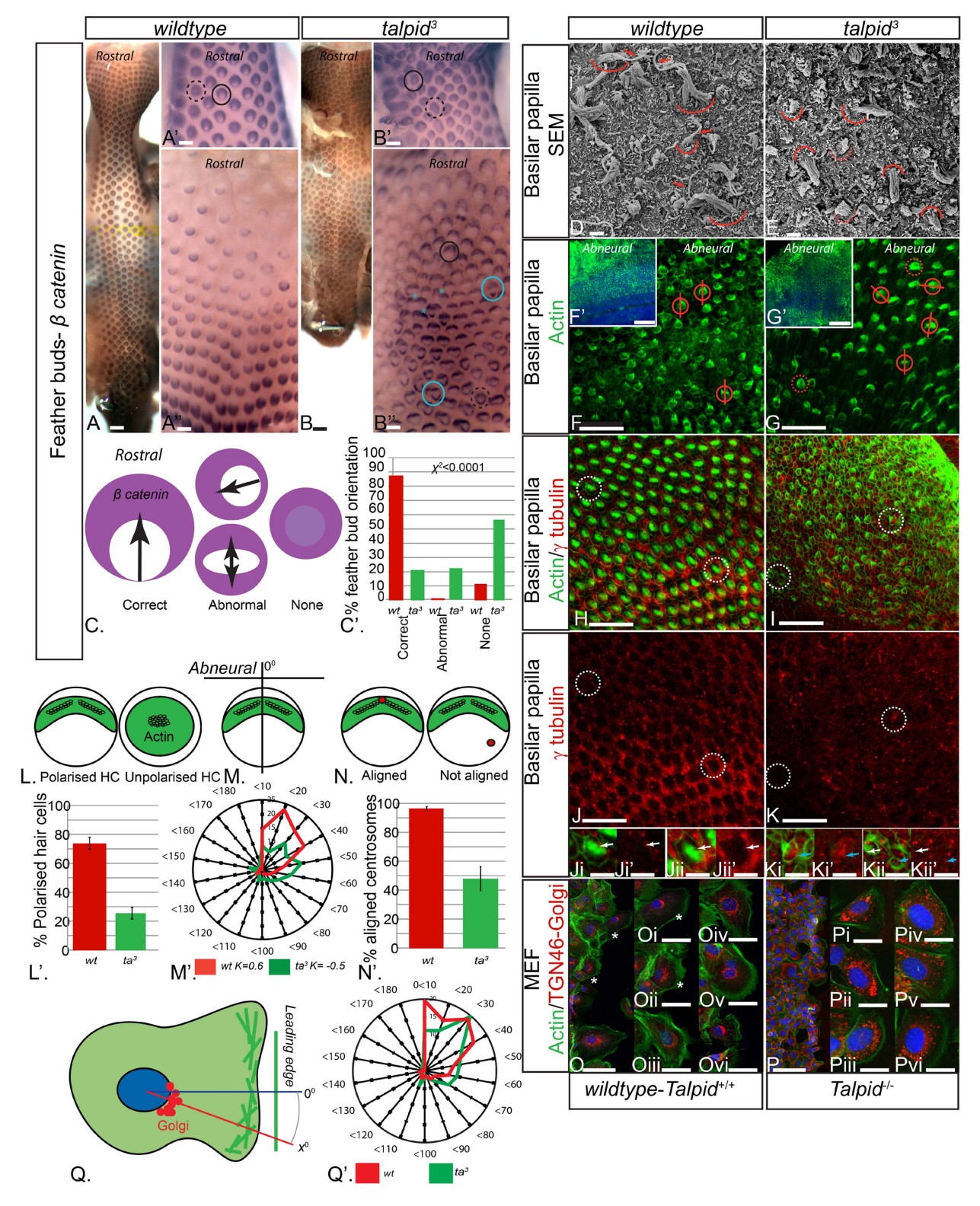

**Figure 3**. Loss of TALPID3 (KIAA0586) causes abnormal tissue and cell polarity and abnormal intracellular organization. (**A**, **A'**, **A''**) *β-catenin* expression is localized anteriorly within feather buds of the *wildtype* chicken at day 9.5 and (**B**, **B'**, **B''**) in the *talpid³* chicken at day 9.5. Black circles indicate feather buds with correct polarity; dashed black circles represent no polarity; blue circles represent abnormal polarity (Schematic **C**). The *talpid³* chicken (**B''**) demonstrates feather buds which lack polarity (blue circles) which is not seen in the *wildtype* chicken (**A''**). Asterisks represent merged feather buds.
*Figure 3 continued on next page*

*Figure 3 Continued*

(**C'**) Quantification of the percentage of feather buds with correct, abnormal or no polarity in *wildtype* and *talpid³* (**D, E**) SEM of the basilar papilla in *wildtype* (**D**) and *talpid³* (**E**) chickens. Arrows indicate cilia. Curved lines represent the base of stereocilia hair bundles. (**F–K**) Actin bundles identified by phalloidin (green) and centriolar localization identified by γ tubulin (red). (**F', G'**) overview of *wildtype* and *talpid³* basilar papilla, higher magnification in (**F, G**), red circles with line represent orientation of polarized actin bundles in basilar papilla; dashed red circles represent unpolarized actin bundles (Schematic **L, M**). (**L'**) Quantification of polarized haircells. (**M'**) Quantification of the angle of polarised hair cells. (**H–K**) Dashed white circles represent magnified images (**Ji–Kii'**). (**Ji–Kii**) White arrows indicate aligned centrosomes; blue arrows indicate unaligned centrosomes (Schematic **N**). (**N'**) Quantification of cells with aligned centrosomes. (**O, P**) Orientation based on placement of Golgi (TGN46, red) in comparison to actin indicating the leading edge (phalloidin, green) and nucleus (Dapi, blue, schematic in **Q**) in MEFs. Asterisks represent areas of higher magnification (not all represented at lower magnification). (**Q'**) Quantification of the angle of orientation of MEF cells in scratch assay. Scale Bars: **A, B** 5 mm; **A', A'', B', B''** 1 mm; **D, E** 1 μm; **F, G, H, I, J, K** 20 μm; **F', G'** 100 nm; **Ji, Ji', Jii, Jii', Ki, Ki', Kii, Kii'** 10 μm; **O, P** 100 μm; **Oi, Oii, Oiii, Oiv, Ov, Ovi, Pi, Pii, Piii, Piv, Pv, Pvi** 25 μm.

(n = 632 from 7 samples, *Figure 3H,J,N*). In contrast, only 49% of *talpid³* cells exhibited a centriole within the bare zone (either abneural or abnormally polarized; n = 219 from 6 samples; *Figure 3I,K,N*), thus demonstrating that the intracellular organization of *talpid³* cells was frequently abnormal. Furthermore, and in agreement with the failure of correct polarization of the stereocilia, centrioles were frequently observed on the neural side of *talpid³* HCs (*Figure 3Kii*). We conclude that failure of centriolar migration in *talpid³* cells results in abnormal cell polarization and stereocilia formation in HCs. Because 49% of *talpid³* cells did exhibit a centriole correctly localized yet ciliogenesis was completely disrupted, the failure of ciliogenesis may not only be due to impaired centriolar migration.

Directional cell migration is also intimately linked to the localization of the centrosome between the leading edge of the migrating cell and the Golgi apparatus. *Talpid3⁻ᐟ⁻* MEFs show abnormal cell migration (*Bangs et al., 2011*), and we therefore examined if the orientation of the Golgi apparatus to the leading edge of migrating cells was also disrupted by a loss of Talpid3 function in mouse, in an in vitro scratch assay (*Figure 3O,P*). The angle between the leading edge and Golgi was taken as the angle of orientation, with an angle of 0° suggesting perfect alignment of the Golgi to the leading edge of the migrating cell (*Figure 3Q*). The angle of orientation was within 40° in 69% of *wildtype* cells and 55% of *Talpid3⁻ᐟ⁻* cells, whilst 20% of *Talpid3⁻ᐟ⁻* cells exhibited orientation angles greater than 60° compared to 11% of *wildtype* cells (*Figure 3Q'*; *wildtype* cells = 132, *Talpid3⁻ᐟ⁻* cells = 117 from two experiments; *Figure 3O–Q*), suggesting a reduction in intracellular polarization of the Golgi apparatus to the leading edge in the *Talpid3⁻ᐟ⁻* MEFs (*Figure 3O,P*). Thus, KIAA0586 (TALPID3; Talpid3) plays an essential role in the internal organization and polarization of cells, likely through its action on the centrosome.

## Abnormalities of intracellular organization, centriole maturation, and centriolar satellite dispersal in the neuroepithelium

JBTS primarily affects the brain of the patients. The choroid plexus is a highly polarized multiciliated neuroepithelium in which we have previously shown, as now in HCs, a failure of centrosome migration in *talpid³* mutant chickens (*Stephen et al., 2013*). To determine if *talpid³* mutant neuroepithelia exhibit cell polarity defects, we examined the intracellular organization of choroid plexus cells in E8 *talpid³* mutant chickens. *Wildtype* choroid plexus cells exhibited a distinctive polarization with an apical, centriolar zone (CZ, *Figure 4A*) above a separate zone of mitochondria (MZ, *Figure 4A*); the most apical mitochondria were found an average of 7 μm from the apical surface (*Figure 4C*). In contrast, the mitochondria in *talpid³* choroid plexus are found in the most apical zone, an average of 3 μm from the apical surface (m, *Figure 4B*), and centrioles are present throughout the cell (asterisk in *Figure 4B*). We conclude that the neuroepithelium has an abnormal intracellular organization of centrosomes and mitochondria and therefore, like the HCs and migratory fibroblasts, is not correctly polarized. Although we have previously suggested that a failure of centrosome migration to the apical surface is the primary reason that cilia fail to form (*Stephen et al., 2013*), our analysis of the HCs suggest an additional requirement for TALPID3 during ciliogenesis, independent of the centriole migration. We therefore investigated the maturation of the mother centriole, crucial for the basal body to dock to the membrane and initiate ciliogenesis. Subdistal appendages were identified in approximately 40% of *wildtype* and *talpid³* centrioles (*wt* n = 35, *talpid³* n = 48, *Figure 4D,E,G*),

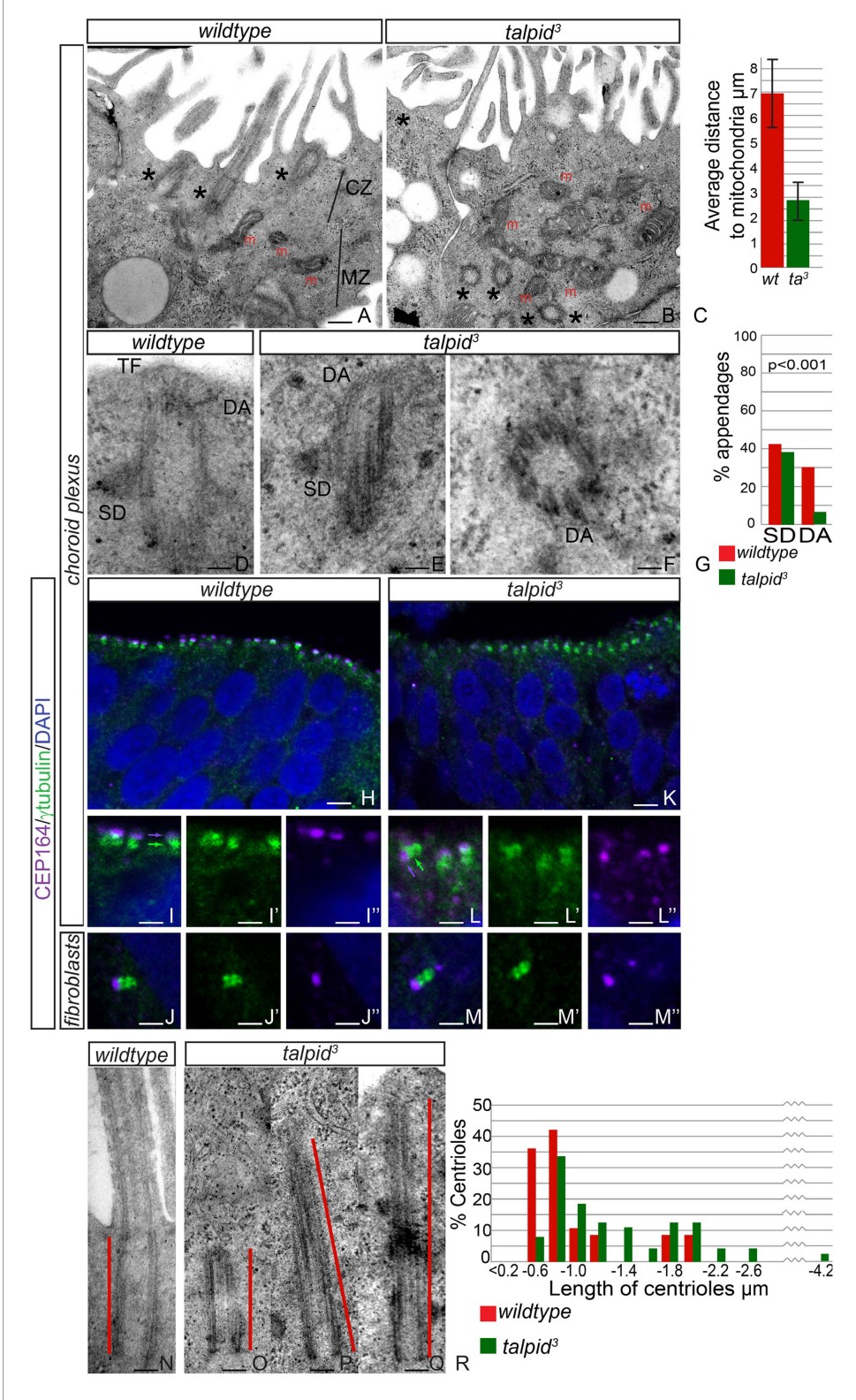

**Figure 4**. Loss of TALPID3 causes abnormal intracellular organization and centriolar orientation (**A**, **B**) The chicken choroid plexus at E8 is a highly polarized structure with docked centrioles (asterisk, **A**) identified within a clear centriolar zone apically (CZ, **A**) and a mitochondrial zone (MZ; m indicates mitochondria). The *talpid³* choroid plexus (**B**) lacks these defined zones, with mitochondria identified in the most apical zone (m, **B**) centrioles identified
*Figure 4 continued on next page*

*Figure 4 Continued*

throughout the cell, failing to dock (asterisk, **B**). Quantification of distance of mitochondria to cell surface (**C**). (**D–G**) *talpid³* tissue is capable of producing mature centrioles. *Wildtype* centrioles (**D**) and *talpid³* centrioles (**E**, **F**) exhibited subdistal appendages (SD), and distal appendages (DA), although DA were less frequently observed on *talpid³* centrioles, quantified in (**G**). CEP164 localizes to the distal mother centriole in *wildtype* and *talpid³* choroid plexus neuroepithelium (purple arrow indicated distal mother centriole, green arrow proximal centriole; **H**, **I**, **I'**, **I''**, **K**, **L**, **L'**, **L''**) and fibroblasts (**J**, **J'**, **J''**, **M**, **M'**, **M''**), but CEP164 puncta are smaller and disorganized in *talpid³* choroid plexus and fail to orientate to the apical surface of the cell (arrows **L**). Centrioles in *wildtype* tissue were on average 0.7 µm (red line indicating centriole/basal body; **N**, **R**) compared to 0.9 µm in the *talpid³* choroid plexus (**O**, **P**, **Q**, **R**). Scale bars: **A**, **B** = 1 µm, **D**, **E**, **F** = 100 nm; **H**, **K** = 10 µm **I**, **J**, **L**, **M** = 5 µm, **N**, **O**, **P**, **Q** = 200 nm.

whereas distal appendages were noted in 28% of *wildtype* centrioles and only 6% of *talpid³* centrioles (*Figure 4D–G*). To determine if there was a loss of distal appendages, we examined localization of CEP164, a protein known to localize to the distal appendages of the mature mother centriole, the basal body. CEP164 localized correctly at the mother centriole and not at the sister centriole, in both *wildtype* and *talpid³* cells of the neuroepithelium and fibroblasts (*Figure 4H–M*). However, CEP164 puncta were smaller, disorganized and frequently orientated away from the apical cell surface in *talpid³* cells (*Figure 4K,L*). This confirmed our previous EM analysis (*Yin et al., 2009*) and data in this study, which demonstrated that centrioles frequently failed to migrate or orientate correctly in *talpid³* cells. Smaller sized CEP164 puncta also suggested that distal appendages were not formed normally in *talpid³* cells. As abnormal or absent distal appendages can result in elongation of the centriole due to improper capping, centriolar length was studied in *wildtype* and *talpid³* choroid plexus cells (*Figure 4N–Q*). Centrioles in *wildtype* tissue were on average 0.7 µm in length compared to 0.9 µm in the *talpid³* chicken, suggesting that *talpid³* centrioles may indeed fail to undergo complete maturation and are subsequently elongated (*Figure 4R*).

In human cells, KIAA0586 is also required for centriolar satellite dispersal (*Kobayashi et al., 2014a*). Compatible with this, we observed electron-dense condensations around the centrioles in the neuroepithelium of *talpid³* chicken, which were absent from *wildtype* centrioles (basal body; 80% of wildtype cell exhibited electron-dense clear area around the centriole, whereas only 21% of *talpid³* cells did; *wt* n = 35, *talpid³* n = 48; *Figure 5A,D,G*). To determine if these were centriolar satellites, we examined the localization of PCM1, a marker for centriolar satellites. Compared to *wildtype* centrioles (*Figure 5B,C*), PCM1 puncta were larger around *talpid³* centrioles (*Figure 5E,F*), possibly reflecting an increase in centriolar satellites. Because we observed KIAA0586 immunostaining around the pericentriolar region (*Figure 2G,H*), we used the centriolar satellite marker AZI1 in human RPE1 cells to determine if KIAA0586 localized to centriolar satellites (*Figure 5H–J*) but found that KIAA0586 and AZI1 did not colocalize. Thus, as observed in human cell lines, TALPID3 is essential for centriolar satellite dispersal. As TALPID3 protein does not localize to the centriolar satellites, we assume that this is an indirect consequence of TALPID3 deficiency.

We conclude that KIAA0586 (TALPID3) is essential for several distinct roles in centriole function, including centriole migration and orientation which can subsequently affect cell and tissue polarity and ciliogenesis, centriole maturation which affects docking of the basal body and ciliogenesis and through an indirect mechanism, centriolar satellite dispersal, which may also affect ciliogenesis.

## Discussion

JBTS is a genetically heterogeneous condition, caused by mutations in several genes related to the structure and function of cilia (*Romani et al., 2013*). Through homozygosity mapping and WES, we identified a novel disease locus (*JBTS23*), defined by mutations in the *KIAA0586* gene, encoding a centrosomal protein (*Andersen et al., 2003*) (*Figure 1*), which is supported by simultaneous concurrent studies (*Bachmann-Gagescu et al., 2015*; *Roosing et al., 2015*). We used *Talpid3/TALPID3⁻/⁻* mouse and chicken models to understand the corresponding pathomechanisms causing the phenotypes of these patients and discovered centrosome abnormalities and loss of cell polarity.

We confirm localization of KIAA0586 at centrosomal structures at the basal bodies and the adjacent daughter centrioles of primary cilia of mouse and human photoreceptor cells as well as in

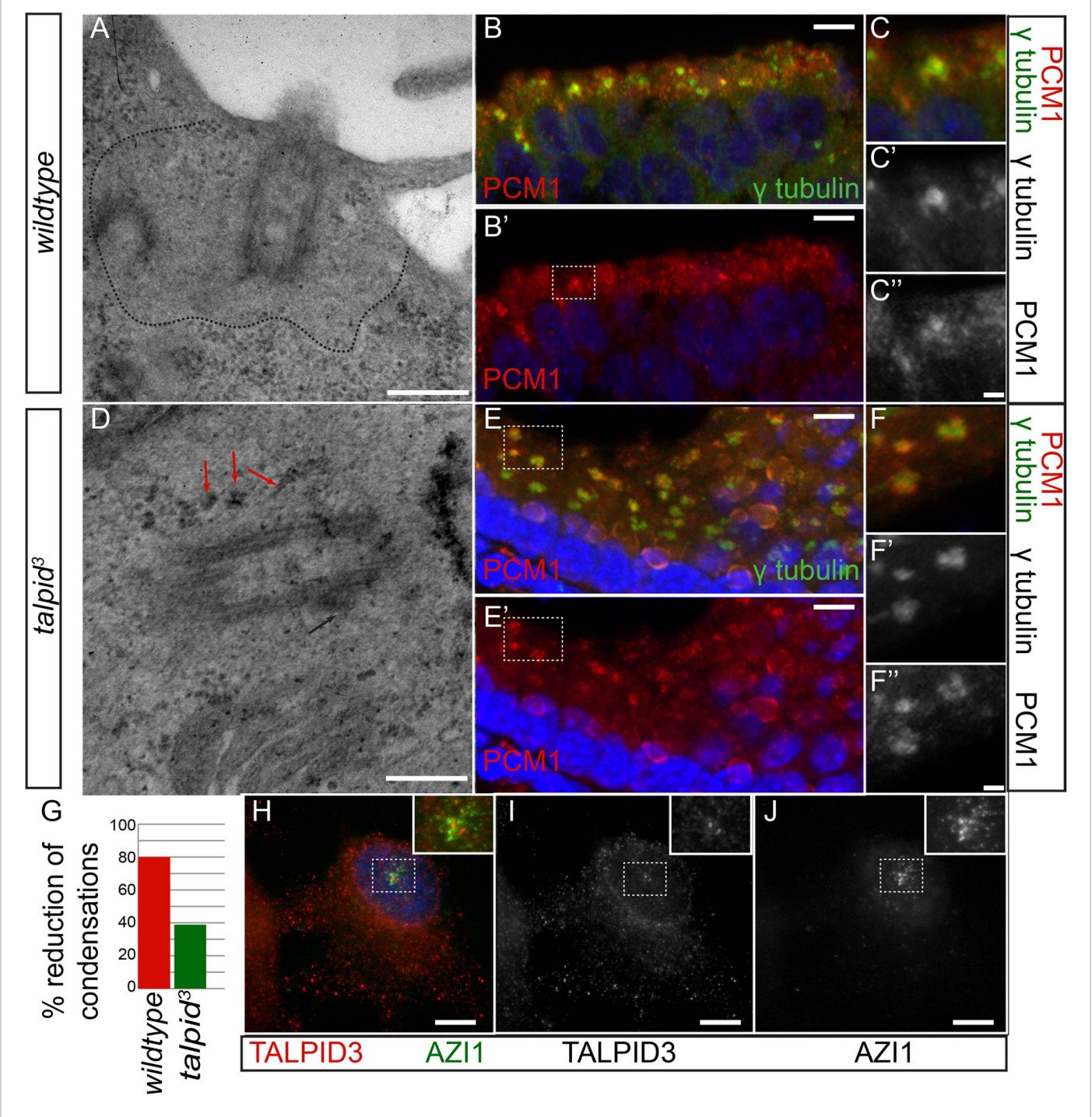

**Figure 5**. Analysis of centriolar satellites in the *talpid³* choroid plexus. An area clear of electron-dense condensations was observed around the basal body in *wildtype* cells (area outlined by dots; **A**), electron-dense condensations were observed adjacent to *talpid³* centrioles (indicated by arrows, **D**). Quantified in (**G**). Immunostaining for a centriolar satellite marker in the choroid plexus, PCM1 (magnified area outlined by dashed line; PCM1 = red, γ tubulin, green **B**, **B'**, **C**, **C'**, **C''**, **E**, **E'**, **F**, **F'**, **F''**). KIAA0586 protein does not colocalize with AZI1, a satellite protein in human RPE1 cells (KIAA0586 = red, AZI1 = green **H**, **I**, **J**). Scale bars: **A**, **D** = 500 nm; **B**, **E** 10 = μm; **C**, **F** = 2 μm **H**, **I**, **J** 5 μm.

pericentriolar regions (*Figure 2*). KIAA0586 has previously been associated with recessive ciliopathy phenotypes in mouse (*Bangs et al., 2011*; *Davey et al., 2014*), chicken (*Davey et al., 2006*; *Davey et al., 2007*) and zebrafish (*Ben et al., 2011*). These animal models have either naturally occurring or

induced 5′ mutations which disrupt an essential coiled-coil domain, resulting in loss of protein function, consecutive loss of Hh signaling and early embryonic lethality. The talpid³ chicken is a thoroughly examined animal model with polydactyly, holoprosencephaly, abnormal neural tube patterning, polycystic kidneys, liver fibrosis, short ribs, and endochondral bones with defective ossification (*Lewis et al., 1999*; *Buxton et al., 2004*; *Davey et al., 2006*; *Davey et al., 2014*).

The c.428delG (p.Arg143Lysfs*4) mutation was identified in heterozygous state in patient MD1, *in trans* to a nonsense mutation (*Figure 1B*), and in a patient G2 who is also heterozygous for a *KIF7* (*JBTS12*) frameshift mutation and variants in three other known ciliopathy genes (*Figure 1C*) (*Dafinger et al., 2011*). Our experiments did not indicate epistatic interaction between *KIAA0586* and *KIF7*, and a secondary occult mutation in either gene cannot be excluded. The c.428delG mutation results in a premature termination codon in five human *KIAA0586* isoforms, causing either a major protein truncation 5′ to the essential coiled-coil domain or NMD. It is comparable to the talpid³ chicken loss-of-function mutation which introduces a frameshift in the orthologous region (*Figure 1F*). The c.428delG mutation is annotated in dbSNP (rs534542684), and its MAF in the general population is surprisingly high (0.39%, 378 out of 96,534 alleles in the ExAC db), reminiscent of the most common deafness (c.35delG in *GJB2*; 0.60% in the ExAC database) or cystic fibrosis (p.Phe508del in *CFTR*; 0.67% in the ExAC database) mutation. In two concurrent studies reporting *KIAA0586* mutations in patients with JBTS, c.428delG represented the most prevalent mutation (*Bachmann-Gagescu et al., 2015*; *Roosing et al., 2015*). While c.428delG was clearly enriched in patients with biallelic *KIAA0586* mutations in both studies (present in 20 of 24 such patients), only two were homozygous. Despite its commonness, c.428delG was neither observed in homozygous state in healthy individuals in the TGP, ESP, or ExAC databases. Such rarity of homozygosity could indicate that it causes embryonic lethality, early death or severe illness leading to underrepresentation of the respective samples. Embryonic lethality in talpid³ chicken and Talpid3⁻/⁻ knockout mice would support such an interpretation. On the other hand, c.428delG was not found in a simultaneous study that reports biallelic *KIAA0586* mutations in early lethal ciliopathies (*Alby et al., 2015*). Of note, a very recent study on rare human knockouts identified in the genomes of 2636 healthy Icelanders lists one individual of 57 years with homozygosity for c.428delG (*Sulem et al., 2015*). This could either be due to protective modifiers or a low mutational load in the ciliome of the respective person. Assuming the latter, c.428delG$_{KIAA0586}$ could represent a hypomorphic allele that increases susceptibility to develop JBTS, with more severe mutations required either *in trans* (in heterozygous carriers, as in most patients reported by *Bachmann-Gagescu et al. (2015)*; *Roosing et al. (2015)*), or in other genes (in homozygous carriers) for disease manifestation. The presence of a heterozygous potentially deleterious *C5orf42* (*JBTS17*) variant in the only c.428delG$_{KIAA0586}$-homozygous patient reported by *Bachmann-Gagescu et al. (2015)*, and the co-occurence of such variants in four ciliopathy genes in patient G2 (including a truncation in the JBTS gene *KIF7*) support the categorization of c.428delG$_{KIAA0586}$ as a hypomorphic mutation of incomplete penetrance. Of note, no secondary *KIAA0586* mutation was identified in c.428delG-heterozygous JBTS patients in the two other studies (*Bachmann-Gagescu et al., 2015*; *Roosing et al., 2015*), which could be due to the contribution of other genes.

The homozygous mutations c.2414-1G>C (Family 1) and c.2512C>T (p.Arg838*, Family 2) would not disrupt the 5′ functionally essential coiled-coil domain in the consecutive KIAA0586 protein, and partial function may be maintained (possibly due to preserved, albeit truncated, KIAA0586 protein). We have shown that KIAA0586 has several functions in the centriole, and this may be mediated by different protein residues.

The occurrence of retinal degeneration in JBTS depends on the genetic subtype, but is variable even within a family. The localization of KIAA0586 at the ciliary base of retinal photoreceptor cells corresponds to other JBTS proteins. Proteins of the periciliary compartment at the base of the photoreceptor cilium are thought to be critical for the handover of cargo from the dynein-mediated transport through the inner segment to the kinesin-powered anterograde intraflagellar transport in the ciliary compartment (*Roepman and Wolfrum, 2007*; *Sedmak and Wolfrum, 2010*). KIAA0586 may be part of the protein networks implicated in these processes. The lack of retinal disease in the patients described herein may be due to the less strongly developed structure of the distal appendages and/or the possible functional redundancy in the cilia of retinal photoreceptor cells. Nevertheless, patients with *KIAA0586*-related JBTS should be investigated for signs of retinal degeneration, and given that mutations in the JBTS gene *CEP290* may cause non-syndromic Leber

congenital amaurosis (*den Hollander et al., 2006*), *KIAA0586* represents a candidate gene for isolated retinopathies.

Loss of KIAA0586 (TALPID3) function in animal models results in a failure to produce both primary and motile cilia. Previously, it has been suggested that this is due to a failure of the centrosome to migrate apically or dock at the plasma membrane (*Yin et al., 2009*; *Stephen et al., 2013*). The subsequent failure of cilia formation results in abnormal Hh signaling and disrupted GLI processing (*Davey et al., 2006*). Most patients with *KIAA0586*-related JBTS exhibit few classical Hh phenotypes such as polydactyly (this study, *Bachmann-Gagescu et al., 2015*; *Roosing et al., 2015*), unlike the corresponding mouse, chicken, and zebrafish models (*Davey et al., 2006*; *Bangs et al., 2011*; *Ben et al., 2011*). We show that, independent of Hh signaling, cell and tissue polarity are disrupted upon loss of *TALPID3*. JBTS is characterized by cerebellar hypoplasia and loss of decussation of neuronal projections from the cerebellum (*Romani et al., 2013*). While Hh signaling is required for controlling the growth of the embryonic cerebellar primordia (*Lewis et al., 2004*), the failure of decussation has been proposed to result from defective axonal guidance (*Romani et al., 2013*), a process depending on centrosome-guided cell polarity (*Solecki et al., 2006*). Furthermore, we have shown that in inner ear HCs, cell polarity and ciliogenesis, albeit closely linked, are differentially affected in *talpid³* cells. Thus, loss of decussation may reflect loss of polarity.

We propose that KIAA0586 exerts a role in intracellular trafficking and cell polarity distinct from its role in docking of the centriole. *Talpid³* cells have abnormal microtubule dynamics (*Yin et al., 2009*). Microtubules are required for the recruitment of satellites and proteins in the distal centriole (*Kim et al., 2008*; *Schmidt et al., 2009*), a process known to be impaired by loss of KIAA0586. Abnormal cell polarity in *talpid³* cells may be due to the effect of TALPID3 on microtubule dynamics and a direct role in centrosome organization: microtubules are essential for intracellular trafficking, cellular structure, and polarity. We have shown that localization of the centrosome, mitochondria, and Golgi is disrupted in *talpid³* cells. Moreover, Rab8, a GTPase which binds to the Golgi and is required for vesicular trafficking and ciliogenesis (*Nachury et al., 2007*; *Henry and Sheff, 2008*; *Feng et al., 2012*), is mislocalized in *KIAA0586*-depleted cells (*Kobayashi et al., 2014a*). In JBTS patients with mutations in *AHI1* (*JBTS3*), encoding an interactor of RAB8 (*Hsiao et al., 2009*), non-ciliary trafficking from the Golgi and ciliogenesis are impaired. Of note, Golgi mislocalization in the *talpid³* choroid plexus is similar to what has been observed in *Ahi1⁻/⁻* mice (*Hsiao et al., 2009*). This suggests a similar pathogenesis of *JBTS23* and *JBTS3*, with defective cell polarity, intracellular trafficking, and Hh signaling.

KIAA0586 interacts with CP110 and Cep120 (*Kobayashi et al., 2014a*), distal centriolar proteins implicated in centriole duplication and maturation, and ciliogenesis. The predominant expression of Cep120 on the daughter centriole throughout most of the cell cycle depends on Kiaa0586, as indicated by high expression of Cep120 on both centrioles and absence of CP110 from the mother centriole prior to ciliogenesis in *Talpid3⁻/⁻* cells (*Spektor et al., 2007*). Although equal expression of KIAA0586 on the mother and daughter centrioles has been reported (this study, *Kobayashi et al., 2014a*); there is evidence that KIAA0586 predominantly localizes at the mother centriole (*Wu et al., 2014*). In addition, loss of chicken *KIAA0586* (*TALPID3*) causes centriole elongation whereas overexpression of Cep120 causes elongation of the mother centriole, suggesting that KIAA0586 (TALPID3) may control centriole length through depletion or suppression of Cep120 on the mother centriole. Similarly, depletion of CP110 also increases centriole length (*Schmidt et al., 2009*), suggesting that KIAA0586 regulates centriolar length through controlling CP110 localization and centriolar capping of the distal mother centriole. Loss of other centriolar proteins, such as OFD1, likewise results in elongated centrioles and loss of distal appendages (*Singla et al., 2010*). Based on the colocalization of KIAA0586, CP110, and Cep164, it has been proposed that KIAA0586 regulates ciliary vesicle docking adjacent to Cep164 localization (*Kobayashi et al., 2014a*), but not distal appendage formation itself, and this is supported by evidence from human patient *KIAA0586⁻/⁻* cells which show Cep164 within the distal centriole (*Alby et al., 2015*). We also find evidence for a vesicle docking defect, demonstrated by an increase in centriolar satellites. However, we propose that *KIAA0586* loss primarily causes abnormal distal appendages and impaired Cep164 localization, similar to what can be observed in OFD1 mutants (*Singla et al., 2010*). In addition, determination of Cep164 expression in cells of highly polarized tissue demonstrates a further centriolar defect not easily distinguished in in vitro assays—the loss of centriole orientation to the apical membrane of the cell. Whether this defect is due to the

depletion of KIAA0586 from the centriole or impairment of another function of KIAA0586 in pericentriolar regions or cytoskeleton remains to be elucidated.

We have identified *KIAA0586* as a novel gene for JBTS, and we propose that it is not only required for ciliogenesis, but also to establish cell and thus tissue polarity. *JBTS23*, and possibly other JBTS subtypes, may result from impairment of both functions.

## Materials and methods

### Patients

Blood samples for DNA extraction were obtained with written informed consent. All investigations were conducted according to the Declaration of Helsinki, and the study was approved by the institutional review board of the Ethics Committees of the University of Erlangen-Nürnberg, the University of Bonn, and the University Hospital of Cologne.

### Genetic analysis of human JBTS families

In accordance with the Human Gene Nomenclature Committee (HGNC), we have used *KIAA0586*/ KIAA0586 for designation of the human gene and protein, respectively. In accordance with the Chicken Gene Nomenclature Committee (CGNC), we use *TALPID3*/TALPID3 for designation of the chicken gene and protein, respectively. Although the current gene symbol for the mouse gene is *2700049A03Rik* (protein: 2700049A03RIK), we use *Talpid3*/Talpid3 as the gene and protein names, respectively. Where we refer to a generic conclusion on the function of the orthologs of *KIAA0586*, we use *KIAA0586*. As in previous publications, the chicken model is referred to as *talpid*[3], and the mouse model is referred to as *Talpid3*[−/−]. The nomenclature of human *KIAA0586* mutations refers to reference sequence NM_001244189.1 (corresponding protein: NP_001231118.1). The Exome Aggregation Consortium (ExAC) database (Cambridge, MA, United States; http://exac.broadinstitute.org), which aggregates numerous databases including the current versions of the Exome sequencing project (ESP, *Fu et al., 2013*) and the Thousand Genomes Project (TGP, *Via et al., 2010*) was last accessed on 11 July 2015 for the presence and frequency of identified variants in healthy individuals.

Family 1: genotyping and homozygosity mapping were performed in Family 1 (MR026) as previously reported (*Abou Jamra et al., 2011*). DNA from patient MR026-01 underwent exome capture and whole-exome sequencing (WES) using the SureSelect Human All Exon 50 Mb Kit (Agilent Technologies, Santa Clara, United States) and a SOLiD4 instrument (Life Technologies, Carlsbad, United States) as described previously (*Abou Jamra et al., 2011*). Of the targeted regions, 73.2% were covered at least 20×, and 83.4% were covered at least 5×. To validate the results, we also conducted WES in the likewise affected sibling, MR026-04, analogous to previously described disease gene identification approaches (*Ahmed et al., 2015*; *Riecken et al., 2015*). 96% of the target sequence were covered at least 20×.

Family 2: samples from the index patient, MD1, and her parents underwent WES at GeneDX (Gaithersburg, MD, United States).

Family 3: WES and mapping of reads for the index patient (G2) and both parents were carried out as previously described (*Basmanav et al., 2014*; *Beck et al., 2014*). In brief, filtering and variant prioritization was performed using the varbank database and analysis tool (https://varbank.ccg.uni-koeln.de) of the Cologne Center for Genomics. In particular, we filtered for high-quality (coverage >15-fold; phred-scaled quality >25), rare (MAF [minor allele frequency] ≤0.01) variants (dbSNP build 135, the 1000 Genomes database build 20110521, and the public Exome Variant Server, NHLBI Exome Sequencing Project, Seattle, build ESP6500). To exclude pipeline-related artifacts (MAF ≤ 0.01), we filtered against variants from in-house WES data sets from 511 patients with epilepsy. The Affymetrix genome-wide Human SNP Array 6.0 utilizing more than 906,600 SNPs and more than 946,000 copy number probes was used for genome-wide detection of copy number variations in patient G2. Quantitative data analyses were performed with GTC 3.0.1 (Affymetrix Genotyping Console) using HapMap270 (Affymetrix) as reference file. In the index patient (G2), all coding *KIAA0586* and *KIF7* exons were Sanger-sequenced in search of a second mutation. In addition, we amplified and sequenced all *KIAA0586* exons from cDNA (derived from whole blood mRNA, PAXgene Blood RNA Tube, PreAnalytiX, Hombrechtikon, Switzerland) in search of potential hints of aberrant splicing due to extra-exonic variants. Continuous PCR amplification of *KIF7* exons from whole blood mRNA was not successful. The sample of patient G2 was analyzed by genome-wide CGH (Affymetrix 6.0 SNP array) to exclude structural alterations adjacent to or within *KIAA0586*, *KIF7*, *CEP41*, *KIF14*, or

*WDPCP*. Confirmation of the identified mutations and segregation analyses were carried out by Sanger sequencing.

## RT-PCR

In Family 1, we isolated mRNA using the RNeasy kit (Qiagen, Hilden, Germany) from lymphoblastoid cell lines that have been established based on standard protocols from patients MR026-01 and MR026-04. We transcribed mRNA to cDNA using SuperScriptII reverse transcriptase and random primers (Invitrogen; Van Allen Way Carlsbad, California, United States). To test if the *KIAA0586* mutation c.2414-1G>C impairs splicing, we used two pairs of primers (*KIAA0586*_exprF1, 5′-TCCATCTCCTAAGTCCAGACCAC-3′ and *KIAA0586*_expR1, 5′-TCCAAGTTTGCACAGGAGG-3′, located in exons 16 and 19, and *KIAA0586*_exprF2, 5′-TCAGGTACATTGGAAGGTCATC-3′ and *KIAA0586*_expR2, 5′-AACTGGCGGAAATGGAGG-3′, located in exons 17 and 21; NM_001244189.1) and standard PCR methods. Electrophoresis on standard agarose gel followed by cutting out the DNA bands, purifying the DNA using QIAquick gel extraction kit (QIAgene; Hilden, Germany), and Sanger sequencing were performed.

## Animal models

Eggs were obtained from *talpid³* flock (MG Davey; *talpid³* chicken lines are maintained at the Roslin Institute under UK Home Office license 60/4506 [Dr Paul Hocking], after ethical review). Mice were maintained at the Human Genetics Unit, Western General, Edinburgh, under UK Home Office license PPL 60/4424 (Ian Jackson). The *Talpid3⁺/⁻/Kif7⁺/⁻* line was produced by crossing of the previously described *Talpid3⁺/⁻* knockout mouse line (*Bangs et al., 2011*) and the reported *Kif7⁺/⁻* mouse line (*Cheung et al., 2009*). Animal experiments carried out at the JGU Mainz corresponded to the statement of the Association for Research in Vision and Ophthalmology (ARVO) as to care and use of animals in research. Adult mice were maintained under a 12-hr light–dark cycle, with food and water ad libitum.

## Incubation and dissection of animal models

Chicken eggs from *talpid³* flock were incubated at 38°C until 12 days at the latest, staged as per *Hamburger and Hamilton (1951)*, dissected into cold PBS, and fixed in 4% PFA/PBS. Mouse timed matings were established between *Talpid3⁺/⁻* mice (*Bangs et al., 2011*) and *Kif7⁺/⁻* mice (*Cheung et al., 2009*) and confirmed by vaginal plug. Pregnant females were sacrificed at day 10 of pregnancy for production of mouse embryonic fibroblasts (MEFs). Otherwise between day 12 and 16 of pregnancy and embryos were dissected in cold PBS, decapitated, and fixed immediately in 4% PFA for histological analysis. Pups were sacrificed between 7 and 21 days after birth by lethal injection and brains were dissected into 4% PFA/PBS.

## Chicken and mouse genotyping

Embryos used in comparisons were dissected as family groups and genotyped after analysis. Tissues were collected on dissection, lysed in 10 mM Tris (pH8), 10 mM EDTA (pH 8), 1% SDS, 100 mM NaCl, and 20 mg/ml proteinase K at 55°C overnight before DNA extraction using Manual Phase Lock Gel Tubes (5 Prime) for phenol/chloroform extraction. For chicken *TALPID3*, sequencing primers used were 5′-TCATTTCATTAGCTCTGCCG-3′ (forward) and 5′-CCATCAAACCAACAGCTCAG-3′ (reverse). For mouse *Talpid3*, PCR primers were 5′-TGCCATGCAGGGATCATAGC (forward), 5′-GAGCACAC TGGAGGAAAGC-3′ (reverse) and 5′-GAGACTCTGGCTACTCATCC-3′, 5′-CCTTCAGCAAGAGCTGG GGAC-3′, respectively. For mouse *Kif7*, PCR primers were 5′-CACCACCATGCCTGATAAAAC-3′ (P1 forward), 5′-CTATCCCCAATTCAAAGTAGAC-3′ (P1 reverse), 5′-CCAAATGTGTCAGTTTCATAGC-3′ (P2 forward), 5′-TTCTCACCCAAGCTCTTATCC-3′ (P2 reverse).

## Histology

Fixed samples from mouse brain and chicken legs were embedded in paraffin, sectioned, and stained in haematoxylin and eosin as described previously (*Davey et al., 2014*).

## Wholemount RNA in situ hybridization

Mouse and chicken embryos were rehydrated through a methanol gradient and in situ hybridization carried out for chicken β-catenin (codons 1–127) as previously described (*Nieto et al., 1996*).

## In ovo knockdown of *Kif7* in chicken

The following *Kif7* sequences were targeted for knockdown: Target 1: TTATCGACGAGAACGACCTCAt, Target 2: cATCCAGAACAAAGCGGTGGTG, Target 3: gTCCTCTAACACTAAGAACATT, Target 4: gACAGATGACATAGTCCGTGTG to which 22mer sequences were designed in Genscript and cloned into pRFPRNAiC (*Das et al., 2006*) (Dundee Cell Products, Dundee, United Kingdom). Embryos were electroporated at stage 12HH (as described *Yin et al., 2009*), observed for RFP expression at stage 24HH, fixed, and prepared for sectioning and immunohistochemistry at stage 22HH as below. Tissue from embryos was collected and genotyped.

## Cell culture and immunocytochemistry

MEFs were prepared from E10.5 eviscerated and decapitated embryos. Cells were dissociated in trypsin/versin and maintained to passage 2 as per *Hall et al. (2013)* and serum removed from media for 48 hr to induce ciliogenesis. RPE1 cells (ATCC) were grown in DMEM-F12, 10% FCS Gold, 50 μl hygromycin, 5 ml L-glut. IMCD3 (mouse inner medullary collecting duct cells) cells were grown in DMEM-F12 10% FCS. To induce ciliogenesis, RPE1 and IMCD3 cells were starved in DMEM:F12 or Opti-MEM I (Life Technologies, Carlsbad, California, United States) for 72 hr. Cells were fixed with methanol at −20°C for 2–5 min. After washing in PBS, cells were immunolabeled with polyclonal antibodies against acetylated tubulin (T7451, Sigma-Aldrich, St. Louis, Missouri, United States), pericentrin-2 (sc-28145, Santa Cruz Biotechnology, Dallas, United States), and KIAA0586 (HPA000846, Atlas Antibodies, Stockholm, Sweden) before incubation with appropriate secondary antibodies conjugated to Alexa 488 (A21206, Molecular Probes, Invitrogen), CF 568 (20106-1, Biotrend, Köln, Germany), and CF 640 (20177, Biotrend), and with DAPI (6335.1, Roth, Karlsruhe, Germany).

## Immunohistochemistry

Eyes from a healthy human donor (#199-10; 56 years of age, dissection 29 hr post mortem) were obtained from the Department of Ophthalmology, University Hospital of Mainz, Germany, according to the guidelines of the declaration of Helsinki. After sacrifice, eyeballs from adult C57BL/6J mice were dissected, cryofixed in melting isopentane, cryosectioned and immunostained as previously described (*Overlack et al., 2011*). Cryosections were incubated with monoclonal antibodies to centrin-3 as a molecular marker for the ciliary apparatus of photoreceptor cells as previously characterized (*Trojan et al., 2008*), and polyclonal antibodies against KIAA0586 (Atlas HPA000846). Washed cryosections were incubated with appropriate antibodies conjugated to Alexa 488 (Molecular Probes A21206) and Alexa 568 (Molecular Probes A11031) in PBS with DAPI (Roth 6335.1) to stain the nuclear DNA and mounted in Mowiol 4.88 (Hoechst, Germany). Specimens were analyzed in a Leica DM6000B deconvolution microscope (Leica, Germany). Image contrast was adjusted with Adobe Photoshop CS using different tools including color correction. For section immunocytochemistry on chicken tissue, chicken embryos were dissected into PBS, fixed, sectioned, and stained as described (*Davey et al., 2006*), except for CEP164, in which an antigen retrieval step was undertaken (incubation in 0.1% BME/PBS for 5 min, incubation in 55°C PBS for 4 hr). For bone sections, legs were dissected at E12. For immunocytochemistry, cells were fixed as above. Antibodies were used against: acetylated α-tubulin (Sigma–Aldrich T7451), γ-tubulin (Sigma–Aldrich T5192; T5326), TGN46 (Abcam, Cambridge, United Kingdom, ab16059), PCM1 (Abcam ab72443), AZI1 (kind gift of Jeremy Reiter, UCSF), centrin-3 (*Trojan et al., 2008*), KIAA0586 (Atlas Antibodies, HPA000846, ProteinTech 24421-1-AP), CEP164 (ProteinTech, Manchester, United Kingdom, 22227-1) RFP (Life Technologies R10367), GFP (Life Technologies A-21311), Pax7 (Developmental Studies Hybridoma Bank, Iowa City, United States (DSHB)), ISLET1 (DSHB), NKX2.2 (DSHB), Phalloidin (Life Technologies A12379), Anti-mouse (Life Technologies A11017), anti-rabbit (Life Technologies A21207). Imagining was undertaken on a Zeiss LSM 710 or a Nikon Air confocal microscope or Leica DMLB.

## Conventional transmission electron microscopy

Chicken embryos were dissected into PBS at 8 days of incubation, avoiding contamination with yolk, heads were removed and placed into 4% PFA, 2.5% glutaldehdye in 0.1 M cacodylate buffer. The choroid plexus was immediately removed and placed into fresh fixative (as previous) for 24 hr. Tissue was prepared and visualized for transmission electron microscopy as described previously

(*Davey et al., 2007*), and axoneme/basal body structure was compared to what was observed and reported previously (*Paintrand et al., 1992*).

## Immunoelectron microscopy analysis

Anti-KIAA0586 antibody (Atlas Antibodies, HPA000846) was used for pre-embedding labeling in mouse retinas as previously described (*Maerker et al., 2008*; *Sedmak and Wolfrum, 2010*). Ultrathin sections were cut on a Leica Ultracut S microtome and analyzed with a Tecnai 12 BioTwin transmission electron microscope (FEI, The Netherlands). Images were obtained with a charge-coupled device SIS Megaview3 SCCD camera (Surface Imaging Systems, Herzogenrath, Germany) and processed with Adobe Photoshop CS.

## Cell polarization and cilia length measurements

Angles of proliferation, migration, orientation, and localization were calculated using Axiovision Angle3 software, and cilium length was measured using Zen software (Zeiss, Oberkochen, Germany). Scratch assays were carried out in *wildtype* and *Talpid3$^{-/-}$* MEFs grown to confluence and serum starved (DMEM + 0.5% FCS) for 48 hr with a p10 pipette tip. Medium was then renewed and MEFs incubated for four hours before fixation in ice-cold methanol prior to immunofluorescence. Angles of orientation were then taken as a measurement of the angle from the center of the nucleus, through the center of the leading edge (towards the wound, identified by phalloidin staining for F-actin) and again through the center of the Golgi apparatus (identified by TGN46 antibody staining). Tiled Z stacks of the scratch/wound were analyzed for greater accuracy.

The expected orientation of the stereocilia of the basilar papilla hair cells were taken as being at 90° to the abneural edge of the basilar papilla. The angle of orientation was taken by drawing a line through the cell perpendicular to the abneural edge and a second from the center of the cell, intersecting with both the perpendicular line and center of the actin bundle. The internal angle was taken to be the angle by which cell orientation deviated from the expected. Cilium length was measured using Zen software (Zeiss, Oberkochen, Germany).

## Acknowlegments

We are indebted to the families who participated in our study. We thank Prof. Chi-chung Hui, Department of Molecular Genetics, University of Toronto, for the kind gift of the *Kif7* mouse line, Prof. Andrew Forge for help with basal papilla dissection, John James, CHIPs, University of Dundee, United Kingdom, Maurits Jansen of Edinburgh Preclinical Imaging, University/BHF Centre for Cardiovascular Science, University of Edinburgh for technical help, ESRIC for support with advanced imaging, Dr Denis Headon for the kind gift of the β-catenin probe and Dr Jeremy Reiter for the kind gift of the Azi1 antibody. We thank Elisabeth Sehn, and Gabi Stern-Schneider (both JGU Mainz) for their skillful technical assistance.

## Additional information

### Funding

| Funder | Grant reference | Author |
| --- | --- | --- |
| BBSRC Career Track Fellowship Funding | BB/F024347/1 | Louise A Stephen, Gemma M Davis, Lynn McTeir, Megan G Davey |
| European Community FP7/ 2009/241955 (SYSCILIA) | FP7/2009/241955 | Lars Tebbe, Uwe Wolfrum |
| FAUN-Stiftung | | Uwe Wolfrum |
| Foundation Fighting Blindness (FFB) | | Lars Tebbe, Uwe Wolfrum |
| Imhoff-Stiftung | | Hanno J Bolz |
| Stiftung Auge (Deutsche Ophthalmologische Gesellschaft) | | Hanno J Bolz |

| Funder | Grant reference | Author |
|---|---|---|
| Deutsche Heredo-Ataxie-Gesellschaft | | Hanno J Bolz |
| Deutsche Forschungsgemeinschaft (DFG) | AB393/2-2 | Hasan Tawamie, André Reis, Arif B Ekici, Rami Abou Jamra |
| BBSRC ISPG | | Louise A Stephen, Gemma M Davis, Lynn McTeir, Amy M Fraser, Megan G Davey |
| BBSRC EastBio Studentship | | Amy M Fraser |
| University Hospital of Cologne | | Hanno J Bolz |

The funders had no role in study design, data collection and interpretation, or the decision to submit the work for publication.

## Author contributions

LAS, GMD, LMT, EAH, Conception and design, Acquisition of data, Analysis and interpretation of data, Drafting or revising the article; HT, PN, GN, HT, MT, EB, SU, AMF, PM, Acquisition of data, Analysis and interpretation of data, Drafting or revising the article; LT, OR, Acquisition of data; AR, Conception and design, Drafting or revising the article; ABE, Acquisition of data, Drafting or revising the article; ND, Conception and design, Acquisition of data, Drafting or revising the article, Contributed unpublished essential data or reagents; CC, Acquisition of data, Analysis and interpretation of data, Drafting or revising the article, Contributed unpublished essential data or reagents; UW, MGD, HJB, Conception and design, Acquisition of data, Analysis and interpretation of data, Drafting or revising the article, Contributed unpublished essential data or reagents; RAJ, Clinical examination of patients, Conception and design, Acquisition of data, Analysis and interpretation of data, Drafting or revising the article, Contributed unpublished essential data or reagents

## Author ORCIDs

Louise A Stephen, http://orcid.org/0000-0001-6795-0383

## Ethics

Human subjects: Blood samples for DNA extraction were obtained with written informed consent. All investigations were conducted according to the Declaration of Helsinki, and the study was approved by the institutional review board of the Ethics Committees of the University of Erlangen-Nürnberg, the University of Bonn, and the University Hospital of Cologne.

Animal experimentation: Talpid3 chicken lines are maintained at the Roslin Institute under UK Home Office license 60/4506 [Dr Paul Hocking], after ethical review. Animal experiments carried out at the JGU Mainz corresponded to the statement by the Association for Research in Vision and Ophthalmology (ARVO) as to care and use of animals in research.

# Additional files

## Major datasets

The following previously published datasets were used:

| Author(s) | Year | Dataset title | Dataset ID and/or URL | Database, license, and accessibility information |
|---|---|---|---|---|
| Kobayashi T, Kim S, Lin YC, Inoue T, Dynlacht BD | 2014 | Homo sapiens KIAA0586 (KIAA0586), transcript variant 1, mRNA | http://www.ncbi.nlm.nih.gov/nuccore/NM_001244189.1 | Publicly available at the NCBI Gene Expression Omnibus (Accession no: NM_001244189.1). |
| Kobayashi T, Kim S, Lin YC, Inoue T, Dynlacht BD | 2014 | Homo sapiens KIAA0586 (KIAA0586), transcript variant 3, mRNA | http://www.ncbi.nlm.nih.gov/nuccore/NM_001244191.1 | Publicly available at the NCBI Gene Expression Omnibus (Accession no: NM_001244191.1). |

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
