## [Decision Letter]

Thank you for submitting your work entitled “*TALPID3* is a regulator of cell polarity and mutated in Joubert syndrome (*JBTS24*)” for peer review at *eLife*. Your submission has been favorably evaluated by Stylianos Antonarakis (Senior editor) and two reviewers, one of whom, Harry Dietz, is a member of our Board of Reviewing Editors.

The reviewers have discussed the reviews with one another and the Reviewing editor has drafted this decision to help you prepare a revised submission. As you will see, both reviewers expressed considerable enthusiasm regarding this study and manuscript, but expressed some suggestions and concerns that need to be addressed prior to formal acceptance.

While your manuscript nicely documents that deficiency of TALPID3 causes Joubert syndrome and associates will alterations of cellular polarity, it is essential to provide further comment regarding documented or potential links between mechanism and phenotype and to avoid conclusion that your findings document a direct role for TALPID3 in PCP signaling, as opposed to indirect involvement through regulation of centrosome architecture and migration.

A number of minor points also require your consideration:

1) The details regarding the various aberrant transcripts induced by the splice site mutation are somewhat unclear. Why, for example, is the 108bp deletion expected to result in NMD? Why is the protein shown as truncated for this in-frame deletion, but not for the out-of-frame 188bp deletion (if I am reading Figure 1 correctly)?

2) There seems to be excessive discussion about family 3 despite substantial ambiguity regarding the etiology of disease. I think that this could be downplayed in the text.

3) What is the rational for putting Figure 6 in sixth position when it is discussed before Figure 2? Although no genetic interaction is observed these are important results for discerning the underlying causal mutations in patient G2 and therefore warrant a more detailed explanation in the results.

4) In the second paragraph of the subsection “Loss of TALPID3 disrupts planar cell polarity patterning”, the authors note that *talpid*^*3*^ mutant mice lack columnar chondrocytes in the long bones and state that this is similar to loss of PCP signaling. The long bone phenotype observed in the *talpid*^*3*^ mouse is more akin to Ihh mutants and does not resemble PCP mutants, which are able to form columnar chondrocytes however they are misaligned.

5) Unlike in long bones of the *talpid*^*3*^ mouse, the authors report that columnar chondrocytes are present in the *talpid*^*3*^ chicken leg. Higher magnification images are needed of data given in Figure 3 so it is possible to see this. The authors also report development of a poorly defined bone collar. H and E staining is not sufficient to conclude this, Alizarin red would give a definitive answer.

6) Please give evidence of *talpid*^*3*^ columnar chondrocytes where cell division angle is abnormal. Suggest giving higher magnification image of data in Figure 3. Please provide numbers of cells counted. Graph Figure 3 shows 11% of Wt cells align within 25o of expected angle, while in the Results the authors state that all Wt cells align within 20°.

7) In Figure 3, what are the VANGL2 foci and what is the precedence that this is a marker of polarity in columnar chondrocytes? Please indicate in the figure examples of cells counted as having one or two foci or dispersed VANGL2 expression, higher magnification images may help. Please provide numbers of cells counted and statistical analysis.

8) In Figure 3M and N, Golgi staining hard to see, orientation of leading edge hard to interpret from images shown. Please indicate how orientation of Golgi in relation to the leading edge was measured. Please provide numbers of cells counted and statistical analysis.

9) It is unclear from the data shown in Figure 4 how the conclusions were drawn. We suggest providing higher magnification images clearly marking the axis of the ‘expect orientation’ and how actin bundles are orientated in relation to this in Wt cells but misorientated in *talpid*^*3*^ cells as well as indicating *talpid*^*3*^ cells with actin across the apical cell surface. Give higher magnification of centrosomes localized in bare zones in Wt and not in *talpid*^*3*^. Although actin bundles appear to be orientated differently in *talpid*^*3*^ compared to Wt, γ-tubulin staining is present within bare zones in *talpid*^*3*^ in the images shown. Please provide numbers of cells counted and perform statistical analyses for these experiments.

10) Maturation of the mother centriole to a basal body requires assembly of subdistal and distal appendages. The authors clearly show presence of subdistal appendages in *talpid*^*3*^ mutant centrosomes but evidence of distal appendages given in Figure 5 is ambiguous. Immunofluorescence staining with antibodies against distal appendage proteins such as CEP164 would give a definitive answer as to presence of distal appendages on *talpid*^*3*^ basal bodies.

11) More extensive description of data given in Figure 7 required in the text. Consider quantifying PCM1 staining between Wt and *talpid*^*3*^.

---

## [Author Response]

The reviewers have discussed the reviews with one another and the Reviewing editor has drafted this decision to help you prepare a revised submission. As you will see, both reviewers expressed considerable enthusiasm regarding this study and manuscript, but expressed some suggestions and concerns that need to be addressed prior to formal acceptance.

While your manuscript nicely documents that deficiency of TALPID3 causes Joubert syndrome and associates will alterations of cellular polarity, it is essential to provide further comment regarding documented or potential links between mechanism and phenotype

Our lab has a substantial background in investigating the links between the loss of TALPID3 and the ‘disregulation of Hedgehog signaling’ seen in animal models. Previously, we have been able to associate this phenotype with a loss of ciliogenesis (i.e. mechanism). Our study extends our understanding of the phenotype and mechanism. In animal models, we demonstrate loss of cell polarity (which we believe prevents ciliogenesis and, consecutively, Hedgehog signaling). We have increased the discussion of this area.

The phenotype of the JBTS patients is different from what would be expected from a direct extrapolation from the animal models. We discuss the potential reasons for this phenomenon in our revised manuscript. Based on the work in the current study, we are now conducting a follow-up study to investigate if *TALPID3*-related loss of cell polarity alters axonal projections in the central nervous system, which may underlie the ‘molar tooth sign’ of the patients.

We have reconfigured Figures 3 and 4 into a single Figure 3 so as to progressively highlight the link between the disruption of tissue polarity (i.e. skin) to cell polarity (basilar papilla) to centrosome localization (basilar papilla and in cell migration) in *TALPID3* mutant tissue. We have then examined the centrosome architecture to show abnormal centrosomal length and polarized orientation (using Cep164- newly added) and finally examine pericentriolar abnormalities. We feel that this strengthens the discussion of the potential pathomechanisms in *KIAA0586*-deficient patients (loss of tissue polarity and abnormal Hedgehog signaling) in relation to centriolar polarity.

[…] and to avoid conclusion that your findings document a direct role for TALPID3 in PCP signaling, as opposed to indirect involvement through regulation of centrosome architecture and migration.

We have replaced speculations that TALPID3 (KIAA0586) mutant tissue exhibits PCP defects by with a more generalized statement that TALPID3 is required for normal tissue polarity, cell polarity and correct positioning of centrosomes in polarized cells (as appropriate). In addition, we have removed the work on bone development in which we used the PCP component VANGL2 as a marker of chondrocyte polarity, which we hope will also help avoiding this confusion.

A number of minor points also require your consideration:

*1) The details regarding the various aberrant transcripts induced by the splice site mutation are somewhat unclear. Why, for example, is the 108bp deletion expected to result in NMD? Why is the protein shown as truncated for this in-frame deletion, but not for the out-of-frame 188bp deletion (if I am reading*
Figure 1
*correctly)?*

We agree, and we apologize for the confusion. We have redesigned Figure 1, now with corrected protein schemes for the three aberrant transcripts that result from the splice site mutation. The designation of the splice site mutation has also been corrected into c.2414-1G>C (in the previous version of the manuscript, the designation had erroneously been delineated from another (shorter) *KIAA0586* isoform).

2) There seems to be excessive discussion about family 3 despite substantial ambiguity regarding the etiology of disease. I think that this could be downplayed in the text.

We agree. The patient of Family 3 (patient G2), carries the very common allele c.428delG. We have shortened the discussion on this specific patient, but we have modified and extended the general discussion of the c.428delG allele because of very recent literature which indicates that this mutation represents a highly interesting (and possibly underestimated) class of recessive mutations:

The concurrent study of [48] (*eLife*)(48) and of [6] (Human Mutation) identified this mutation in most of their JBTS patients with biallelic *KIAA0586* mutations – but very rarely in homozygous state.

The c.428delG mutation is listed with high frequency in databases of large-scale sequencing projects (ESP, TGP, ExAC), but not in homozygous state. It was neither found in a recent study that identified biallelic *KIAA0586* mutations in early lethal ciliopathies.

A very recent study about homozygous knockout alleles identified in healthy Icelanders listed one healthy adult with homozygosity for c.428delG. We speculate that c.428delG may represent a hypomorphic allele that requires a more severe *KIAA0586* mutation in *trans*, as is the case in most patients reported by us, Roosing et al. and Bachmann-Gargescu et al., or additional mutations in other (ciliopathy) genes if c.428delG is present in homozygous state (as in case of the homozygous patient reported by Bachmann-Gargescu et al.

We speculate that a similar mechanism could apply to patient G2 from Family 3, with double heterozygosity for truncations in two JBTS genes (*KIF7* and *KIAA0586*) and potentially deleterious variants in other ciliopathy genes. However, we have shortened this part of the Discussion.

Taken together, we think that the unusually common c.428delG allele may – at least in some patients – contribute to oligogenic rather than monogenic JBTS.

*3) What is the rational for putting Figure 6 in sixth position when it is discussed before*
Figure 2*? Although no genetic interaction is observed these are important results for discerning the underlying causal mutations in patient G2 and therefore warrant a more detailed explanation in the results.*

We apologize for this mistake. We should have referred to supplementary figure 1, which is now Figure 1—figure supplement 1 in the revised manuscript.

*4) In the second paragraph of the subsection “Loss of* TALPID*3 disrupts planar cell polarity patterning”, the authors note that* talpid^3^
*mutant mice lack columnar chondrocytes in the long bones and state that this is similar to loss of PCP signaling. The long bone phenotype observed in the* talpid^3^
*mouse is more akin to Ihh mutants and does not resemble PCP mutants, which are able to form columnar chondrocytes however they are misaligned.*

We agree with the reviewer that the *talpid*^*3*^ chicken mutant bone defect may primarily be an IHH defect, as we have described previously in the *talpid3*^-/-^ mouse conditional limb knockout (8). In the respective paper, we conclude that IHH signalling is severely affected in this ciliopathy mutant, based on comparisons with both IHH knockout mice and IFT knockout mice (Haycraft et al., 2007). We would also predict that this is true for the *talpid*^*3*^ chicken mutant – particularly based on PTCH1/PTCH2 RNA in situ hybridization that shows loss of expression of these genes in *talpid*^*3*^ chicken, suggesting that IHH signaling is misregulated in the *talpid*^*3*^ endochondral bones (Davey, unpublished).

We believe that the analysis of the *talpid*^*3*^ chicken endochondral bones has validity in describing the cell polarity defects that we observe in this mutant. However, we agree with the reviewer (Point 4) that we cannot sufficiently distinguish the phenotype caused by the IHH signalling defect from Wnt-driven defects (particularly Wnt5a/b) to conclude that the cell polarity defects are not only due to a loss of IHH signalling.

However, our results indicate phenotypic differences between the IHH^-/-^ mouse and *talpid3*^-/-^ conditional limb knockout; the bones are not only short as in the Ihh^-/-^ mutant mouse, but wider and fused and comparable to chicken in which the PCP pathway is manipulated (Li and Dudley, 2009)or in Wnt5a^-/-^ mice. There is also soft tissue syndactyly, which is common in mutants such as Prickle1^-/-^ mice. Thus, signaling through the Wnt PCP pathway may also been altered. IHH signaling is upstream of Wnt5a expression, an important regulator of cell polarity in the bone (Gao et al., 2011) and it is therefore possible that IHH regulates cell polarity through Wnt5a. This has not been tested, although the IHH^-/-^, Wnt5a^-/-^ mice and other manipulations of the Wnt PCP pathway differ in phenotype an issue which has been raised in other ciliopathy papers (i.e. Haycraft et al., 2007). We agree with the reviewer, that we can at present only conclude that loss of cell polarity in *talpid*^*3*^ chicken endochondral bones is likely dependent on IHH through regulation of Wnt5a. There are also other possibilities beyond of the current IHH and Wnt5a theories; loss of cilia in the developing bone could result in a loss of mechanosensation and consecutive loss of bone polarity. Because we are currently not able to determine if the loss of IHH, Wnt signalling or mechanotransduction through defective ciliogenesis is primarily responsible for the cell polarity defects in the bone, we have removed these results.

*5) Unlike in long bones of the* talpid^3^
*mouse, the authors report that columnar chondrocytes are present in the* talpid^3^
*chicken leg. Higher magnification images are needed of data given in*
Figure 3
*so it is possible to see this. The authors also report development of a poorly defined bone collar. H and E staining is not sufficient to conclude this, Alizarin red would give a definitive answer.*

We have previously published an analysis of the bone mineralization (Macrae et al., 2010) in which we show that Alizarin red, TNAP and PHOSPHO1 are all lost in *talpid*^*3*^ chicken endochondral bones. Data were removed as per above.

*6) Please give evidence of* talpid^3^
*columnar chondrocytes where cell division angle is abnormal. Suggest giving higher magnification image of data in*
Figure 3*. Please provide numbers of cells counted. Graph*
Figure 3
*shows 11% of Wt cells align within 25o of expected angle, while in the Results the authors state that all Wt cells align within 20°.*

We apologize for the ambiguity. Our results, as written were based on 50 bins of degrees, which was more precise (Figure 6). However we increased the bin size in the graph in the submission for size. The data has now been removed.

Author response image 1.**DOI:**
http://dx.doi.org/10.7554/eLife.08077.011

*7) In*
Figure 3*, what are the VANGL2 foci and what is the precedence that this is a marker of polarity in columnar chondrocytes? Please indicate in the figure examples of cells counted as having one or two foci or dispersed VANGL2 expression, higher magnification images may help. Please provide numbers of cells counted and statistical analysis.*

There is little precedence that VANGL2 is a marker for polarity in columnar chondrocytes and very few studies have examined localization of VANGL2 protein in different tissues. Until recently, VANGL2 protein was thought to label the proximal side of hair cells of the mammalian inner ear, and this was used extensively in many studies, whereas it has now been shown that localization is in the distal side of the supporting cells (Ezan et al., 2013). We have demonstrated subcellular localization of VANGL2 protein in any tissue (Stephen et al., 2014). To our knowledge, one other group has demonstrated localization of VANGL2 in the long bones (Supplemental figure 4 of Gao et al., 2011), but they did not thoroughly describe or examine this expression. Both this supplemental figure and our own work agree on the localisation of VANGL2 in the proximal long bones of the limb. This however is very different to the distal localisation shown in the chondrocytes of the digits. Data removed as per above.

Because we employed the same techniques in the bone immunohistochemistry that successfully determined VANGL2 localization in the chicken gastrula, we are confident that the bone localization we observe is real. In addition, we have thoroughly quantitfied this (1213 wildtype cells in 6 samples, 1780 *talpid*^*3*^ cells in 6 samples). Data removed as per above.

Subsequently, we have combined Figure 3 and Figure 4 into a combined Figure 3 “Cellular polarity and centrosome localization is disrupted in KIAA0586 null (TALPID3) tissues” (as per above).

*8) In*
Figure 3*, Golgi staining hard to see, orientation of leading edge hard to interpret from images shown. Please indicate how orientation of Golgi in relation to the leading edge was measured. Please provide numbers of cells counted and statistical analysis.*

We have modified these figures accordingly, improving the resolution and adding additional panels to highlight the findings. We have added a scheme to show how we undertook measurements.

*9) It is unclear from the data shown in*
Figure 4
*how the conclusions were drawn. We suggest providing higher magnification images clearly marking the axis of the ‘expect orientation’ and how actin bundles are orientated in relation to this in Wt cells but misorientated in* talpid^3^
*cells as well as indicating* talpid^3^
*cells with actin across the apical cell surface. Give higher magnification of centrosomes localized in bare zones in Wt and not in* talpid^3^*. Although actin bundles appear to be orientated differently in* talpid^3^
*compared to Wt,*
γ*-tubulin staining is present within bare zones in* talpid^3^
*in the images shown. Please provide numbers of cells counted and perform statistical analyses for these experiments.*

We have modified these figures, adding additional magnifications and annotations to highlight what we have described in the text. We have added a scheme to make these data easier to understand.

*10) Maturation of the mother centriole to a basal body requires assembly of subdistal and distal appendages. The authors clearly show presence of subdistal appendages in* talpid^3^
*mutant centrosomes but evidence of distal appendages given in*
Figure 5
*is ambiguous. Immunofluorescence staining with antibodies against distal appendage proteins such as CEP164 would give a definitive answer as to presence of distal appendages on* talpid^3^
*basal bodies.*

We carried out immunohistochemistry with anti-Ninein, anti-OFD1 and anti-CEP164 antibodies, and have found that the anti-CEP164 antibody reacts very well with chicken tissue. These additional data were very helpful in understanding the orientation of the centriole in *talpid*^*3*^ cells, and we have included them in the revised manuscript.

*11) More extensive description of data given in Figure 7 required in the text. Consider quantifying PCM1 staining between Wt and* talpid^3^*.*

We have revised the description of these data. Unfortunately, we were not able to quantify the PCM1 staining.